# Collective self-caging of active filaments in virtual confinement

Maximilian Kurjahn [1,5], Leila Abbaspour [1,5], Franziska Papenfuß[1], Philip Bittihn [1], Ramin Golestanian [1,2], Benoît Mahault [1] & Stefan Karpitschka [1,3,4]

Motility coupled to responsive behavior is essential for many microorganisms to seek and establish appropriate habitats. One of the simplest possible responses, reversing the direction of motion, is believed to enable filamentous cyanobacteria to form stable aggregates or accumulate in suitable light conditions. Here, we demonstrate that filamentous morphology in combination with responding to light gradients by reversals has consequences far beyond simple accumulation: Entangled aggregates form at the boundaries of illuminated regions, harnessing the boundary to establish local order. We explore how the light pattern, in particular its boundary curvature, impacts aggregation. A minimal mechanistic model of active flexible filaments resembles the experimental findings, thereby revealing the emergent and generic character of these structures. This phenomenon may enable elongated microorganisms to generate adaptive colony architectures in limited habitats or guide the assembly of biomimetic fibrous materials.

Emergent collective behaviors of motile biofilaments are ubiquitous across scales in nature, from cytoskeleton constituents[1–4] to colonies of elongated bacteria[5–10] and even millimeter-sized worms[11,12], with promising applications in synthetic, bio-inspired materials[13–15]. In this context, filamentous cyanobacteria are currently attracting significant attention[9,10,16]. These microorganisms are indeed among the oldest, yet still most abundant phototrophic prokaryotes on Earth, fixing vast amounts of atmospheric carbon by photosynthesis[17,18]. They comprise a wide variety of phenotypes, habitats, and lifestyles, covering benthic and planctonic forms in fresh and salt water, as well as adaptations to extreme conditions like hypersaline lakes or hot springs[17]. Driven by climate change and eutrophication, harmful cyanobacterial blooms are intensifying in recent years[19,20]. Such blooms typically develop when submersed populations grow to a critical density, upon which individual filaments aggregate and eventually rise to the surface of the water body[10,17,21]. The role of tychoplanctic species, i.e., organisms that usually dwell on submersed surfaces but eventually aggregate, detach, and rise to the water surface, remains to be elucidated[17]. Beyond their

major ecological impact[17,22], cyanobacteria are gaining relevance as renewable energy source from photobioreactors or for biochemicals production[18].

Filamentous cyanobacteria are long and flexible[23,24], containing up to several hundred, linearly stacked cells. Many species glide actively along solid surfaces or each other[10], exhibiting photo- and scotophobic responses[25], i.e., reversals of their direction of motion when they experience an increase or decrease in illuminance, respectively. These phenomena are currently not understood in detail[24,26,27] but appear to be crucial for aggregation, which, besides its role in blooming, is believed to provide light exposure regulation[28] and protection from other environmental factors[17]. Yet, the genesis of aggregation remains largely unknown[10], especially for benthic and tychoplanctic species that are usually subject to spatially inhomogeneous illumination[9,29]. In contrast to compact photoresponsive agents like E. coli[30,31] or self-thermophoretic colloids[32], for which light-controlled aggregation is well studied[33,34], filamentous agents generally entail additional emergent phenomena that are currently not well understood[10,35].

[1]Max Planck Institute for Dynamics and Self-Organization (MPI-DS), Göttingen, Germany. [2]Rudolf Peierls Centre for Theoretical Physics, University of Oxford, Oxford, UK. [3]Fachbereich Physik, Universität Konstanz, Konstanz, Germany. [4]Centre for the Advanced Study of Collective Behaviour, Universität Konstanz, Konstanz, Germany. [5]These authors contributed equally: Maximilian Kurjahn, Leila Abbaspour. ✉e-mail: benoit.mahault@ds.mpg.de; stefan.karpitschka@uni-konstanz.de

Here, we subject surface-attached ensembles of gliding filamentous cyanobacteria to compact light patterns and investigate their individual and collective responses (see Fig. 1a, b). We find that motility, scotophobic responses, and filament-filament interactions induce structures beyond simple accumulation: A dense aggregate of reticulated, though predominantly tangentially oriented filaments forms at the boundary of the bright region, i.e., a ring near the edge of an illuminated disk (see Fig. 1d). Using a minimal active polymer model, we corroborate the emergent nature of this ordered pattern, showing that it arises without any explicit aligning interaction with the boundary. We expect the ring self-assembly to be a generic feature of filamentous photoresponsive organisms, allowing them to accumulate and collectively align relative to light patterns in natural and artificial settings. Exploring various illumination patterns, we further demonstrate how such photoresponsive behavior offers strategies for the design and control of frustrated nematic assemblies with distinct topologies, without relying on mechanical templating.

## Results

### Accumulation at light boundaries

The response of the filamentous cyanobacterium *Oscillatoria lutea* SAG 1459-3 to light gradients was observed with the experimental setup sketched in Fig. 1a. *O. lutea* is a tychoplanctic freshwater species that forms sessile and floating colonies[36]. Individual filaments are negatively buoyant, but aggregates may develop gas bubbles, enclosed in a mesh of filaments and secreted slime, that cause

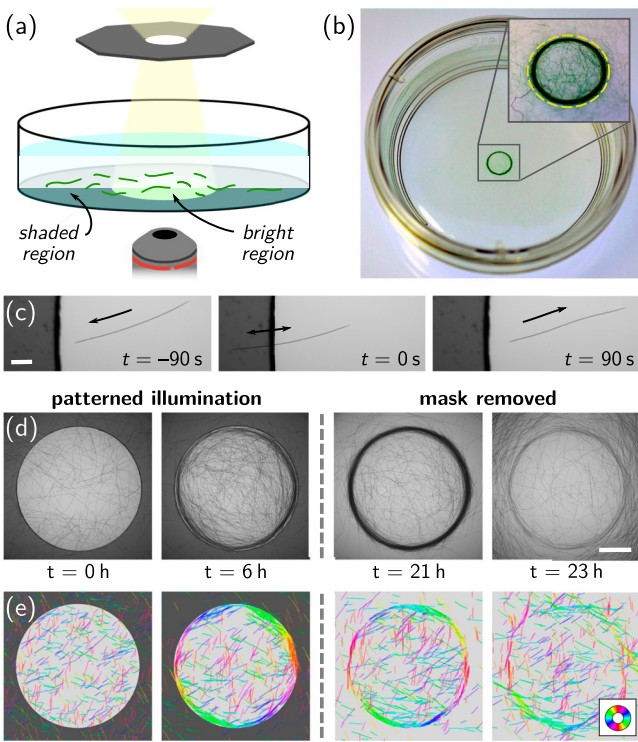

**Fig. 1 | Aggregation of filamentous cyanobacteria at light boundaries.**
**a** Schematic of the experimental setup. Filaments glide at the bottom of a Petri dish, submersed in culture medium and illuminated from above with a pattern generated by the mask. **b** Macroscopic view of a ring structure of *O. lutea* that formed at the edge of a circular light patch (dashed yellow line), far from any physical boundary. The diameter of the Petri dish is 35 mm. **c** Scotophobic response of *O. lutea* when gliding into the dark, scale bar is 100 μm. **d** Formation (*t* = 0–21 h) and dissolution (*t* = 21–23 h) of the ring after removing the mask at *t* = 21 h, scale bar is 1 mm.
**e** Numerical simulations of active filaments (details on the model in the main text) replicate the experimental observations from (**d**). The color wheel indicates the nematic orientation of individual filaments.

flotation. The mechanical properties and collective behavior of *O. lutea* have recently been investigated[9,23,24,37]. Filaments are about 5 μm wide, flexible, and exhibit bi-directional gliding motility that is sustained in complete darkness. Filaments were contained in a Petri dish, filled with liquid medium, and closed by a lid. The sample was illuminated by the bright-field transmission illumination of an inverted microscope, focusing a mask onto the bottom surface, to generate compact light patterns like disks, ellipses, or polygons (see Fig. 1 and "Methods" for a detailed description). In order to also image the outside of the illuminated patterns, the mask was realized by a green filter that reduces the total light intensity by ~80%. Since *O. lutea* is not photosensitive in the green[38], the shaded region appears effectively dark to the filaments. Importantly, there were no other confining objects present in the dish, and both the typical light pattern size (0.5–4 mm) and filament length (*L* ≈ 0.2–2 mm) were much smaller than the Petri dish diameter (≈35 mm). Before starting the illumination, virtually all filaments were attached to the bottom surface of the Petri dish, homogeneously dispersed, and gliding in random directions.

Figure 1c shows the response of individual filaments when they encounter the edge of an illuminated area (see also Supplementary Movie 1). Filaments reverse their gliding direction shortly after a fraction of their body enters the dark region. Importantly, this reversal does not induce any rotation of the filament body, as only the gliding direction is flipped, irrespective of the angle by which the boundary is approached. This behavior is in stark contrast with the mechanical or hydrodynamic interactions of self-propelled particles with solid walls, which usually lead to re-orientations and cause an accumulation of trajectories at boundaries[39–41].

Despite the absence of any physical barrier, illuminating the colony with a circular pattern for 10–20 h leads to the formation of a ring-like structure, where filaments mostly accumulate at and align with the boundary of the virtual confinement (see Figs. 1d, 2a, and Supplementary Movie 2). This structure formation is governed by motility. Growth, in contrast, is much slower (doubling period of the order of days) and cannot be the driving force of the accumulation. The resulting, macroscopically visible aggregate (Fig. 1b, inset, boundary of the bright region indicated with a dashed line) consists of thousands of individual filaments. Its formation is reversible as, upon removing the mask, filaments move away tangentially leading to the progressive dissolution of the ring structure (see Fig. 1d and Supplementary Movie 2).

The aggregation process during the experiment is analyzed in Fig. 2a, showing the azimuthally averaged filament density (top) and nematic order (bottom) as color code versus time and distance *r* to the center of the illuminated area. The local density of bacteria is estimated through their light absorption according to the *Beer-Lambert* law (see "Methods"). The nematic order reflects the degree of alignment of nearby filaments and is derived from the orientation of local variations in the density (see Fig. 2a and "Methods" for details). In the first 3 h, the overall density in the illuminated area increases quite homogeneously, remaining slightly lower near the boundary. After that, however, density develops a peak in a tightly focused region near the boundary of the light spot. The ring structure is mainly fed by filaments that arrive from the outside, but the central region also contributes and thus depletes slightly. After about 10 h, density has reached a stationary state although individual filaments keep moving at the same typical speed (see Supplementary Movie 2). This scenario is further evidenced by Appendix Fig. S2, which shows that the filament densities at the center and edge of the illumination domain grow nearly proportionally at short times, while the formation of the ring is typically associated with a sudden increase of the latter.

As is frequently observed for aligning self-propelled particles in the dilute regime[42], the magnitude of the local nematic order parameter closely follows the local filament density. In the ring structure,

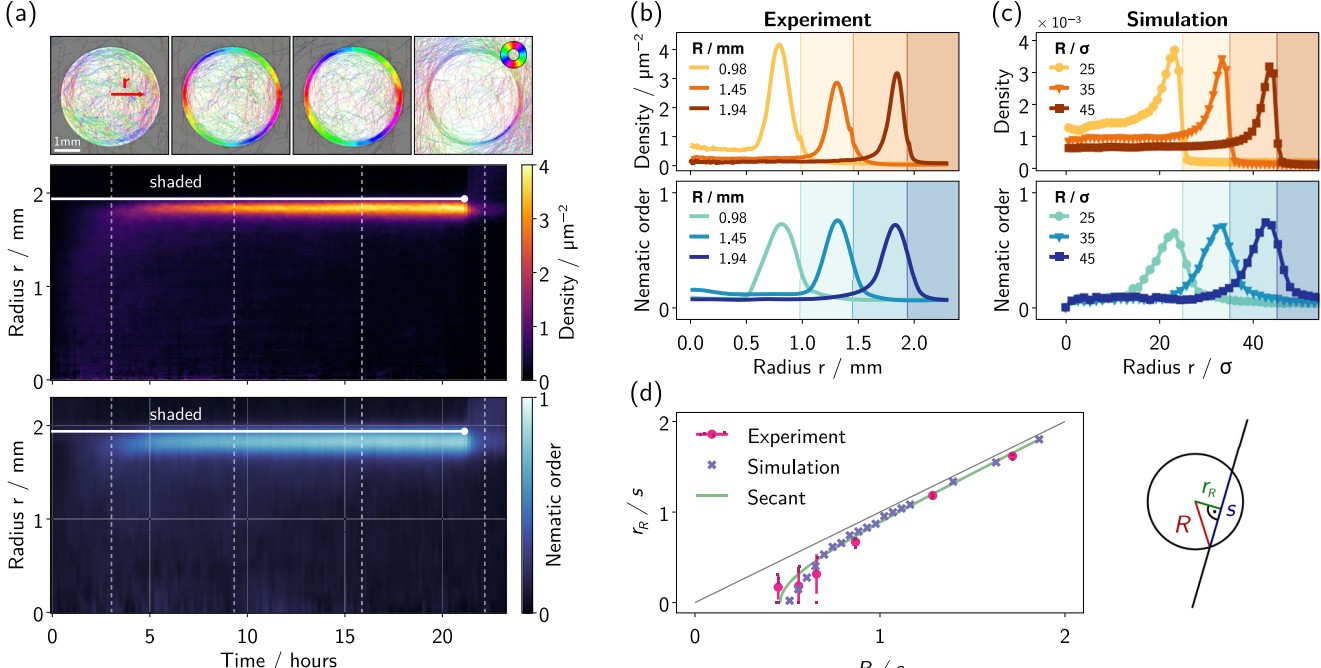

**Fig. 2 | Evolution of the aggregation. a** Local orientation of *O. lutea* at the bottom surface of a Petri dish, illuminated with a circular patch ($R = 1.94$ mm) of white light. Filaments align with and accumulate at the edge of the bright region (see color code), despite the absence of any physical barrier. The mask is removed after 21 h, and the pattern dissolves. The kymographs show azimuthally averaged density and nematic order as function of radius *r* and time. Vertical dashed lines indicate snapshot times, the edge of the light patch is indicated by horizontal white lines where the circle indicates the time point at which the mask was removed. **b** Filament density and nematic order parameter averaged over the stationary time window (-10–21 h) against radius for various mask radii *R*. **c** Numerical simulations of active filaments reproducing the experimental observations (parameters and additional details in "Methods"). **d** Radius of the density peak $r_R$ against mask radius *R*, both normalized by the fitted secant length *s*. The error bars of the experimental data are the standard deviation of 3–8 measurements (individual data shown as small dots). The tangent radius of secants to the masks, with constant secant length *s*, is shown in green (cf. schematic next to the graph). Source data are provided as a Source Data file.

filaments are strongly aligned with the boundary and each other, while the central region does not exhibit strong nematic order. Both, density and nematic order, decrease quickly after the mask is removed, leaving behind only a faint structure of residues that eventually also dissolve.

## Minimal model & mechanism

To confirm that the formation of the ring is not due to some hidden active re-orientation at the boundary, we replicate this collective behavior in simulations of an active filament model (Fig. 1e and Supplementary Movie 3). In the spirit of previous works[24,43–46], filaments are composed of beads connected by springs with typical rest length $\sigma/2$, capable of gliding on the 2D plane and interacting only via repulsion (schematic in Fig. 3b). We use a soft harmonic potential that allows for partial overlap between filaments, reminiscent of the experiments where bacteria may cross each other (compare Fig. 1d). Below, it will turn out that this feature is essential for the collective aggregation at the light boundary. Scotophobic responses are implemented by reversing the gliding direction of the filament once its foremost bead enters the dark. Hence, no torque or rotation of the body is applied. The length distribution of filaments is chosen similar to the experiment (Appendix Fig. S1). The active filament model behaves remarkably similar to the cyanobacteria, showing accumulation at and alignment with the inner boundary of the illuminated disk (color code in Fig. 1e). We find that this behavior is quite robust against changes in the model parameters. Simulation results thus confirm that the ring formation happens without imposing alignment or torques at the boundary, rendering it a purely emergent effect induced by the interactions between the filaments. Kymographs of density and nematic order (see Appendix Figs. S3 & S4) are comparable to the experiments.

To analyze the radial structure of density and nematic alignment in the stationary state, we conducted additional experiments for different mask radii. Radial profiles in Fig. 2b were obtained by averaging density and nematic order over the second half of the patterned illumination phase (-10–21 h). Consistent with the kymographs, both quantities peak around the same position inside of the illuminated region. While the magnitude of the density peak varies with the global filament density for which we have limited experimental control, the peak positions are found to be more robust to varying experimental conditions. In particular, the gap between the mask edge (indicated by the shaded region and visible by a little artifact in the density signal in Fig. 2b) and the peak maximum increases for smaller mask radii *R*, and thus depends on the boundary curvature. Density profiles from our simulations were analyzed the same way (Fig. 2c). Here too, density is maximal somewhat inside of the edge of the illuminated region.

The peak position $r_R$ is estimated by a Gaussian fit, limited to the peak region such that different background densities on either side do not distort the estimation (see "Methods"). In a few experiments with very small radii, no ring but rather homogeneous accumulation occurred (see Appendix Fig. S5), which was taken into account by setting $r_R = 0$. Figure 2d shows $r_R$ as function of the mask radius *R* for experiments (pink disks) and simulations (lavender crosses). The characteristic relation of $r_R$ tending to zero at finite *R* can be explained from the finite size of the filaments. Indeed, the data follows the curve for the inner tangent radius of secants with constant length *s*, given by simple geometric considerations (see sketch in Fig. 2d) as

$$r_R = \sqrt{R^2 - (s/2)^2}. \tag{1}$$

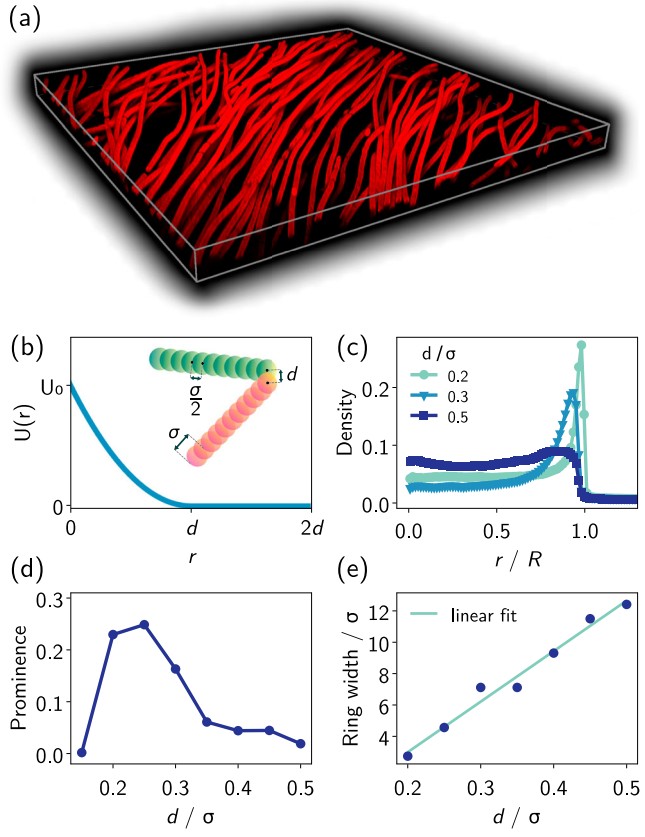

**Fig. 3 | Effects of overlapping filaments. a** 3D rendering of a $z$-stack of 95 confocal images of a part of a ring with mask radius $R = 1.3$ mm. The bounding box dimensions are $442 \times 442 \times 25.4$ μm. Filaments in the ring cross each other, but remain within ~5 layers above the substrate. **b** Harmonic potential function, $U(r)$, governing interactions between beads from distinct filaments. The inset schematic illustrates the parameters $r$ (bead separation) and $d$ (interaction range). **c** Radial distribution of filament density, averaged over time with mask $R = 60\,\sigma$. Three different values of $d$ showcase its impact. **d** Prominence of the ring density (relative height between peak and center density) and (**e**) ring width as a function of $d$, defined from the maxima and half maximum width of the density peaks of (**c**). Source data are provided as a Source Data file.

The secant length $s$ is obtained by fitting Eq. (1) to the data, yielding $s = (1.13 \pm 0.08)$ mm for the experiments, remarkably close to the mean filament length $\bar{L} = 1.11$ mm (see Appendix Fig. S1). Simulations yield $s = (21.5 \pm 0.8)\,\sigma$, while the mean filament length $\approx 20\,\sigma$.

To rationalize this correspondence, we consider filaments that approximately form secants to the boundary. These filaments quickly encounter the dark after gliding only a small distance, triggering frequent reversals that lock them onto short seesaw trajectories. Incoming filaments may then be aligned upon steric interaction. These findings suggest that bending into curved boundaries remains limited, leading to the vanishing of boundary accumulation at a finite domain radius. The absence of bending is a consequence of the virtual confinement: Although self-propulsion forces are sufficient to induce significant bending[24], there is no physical wall to counter the thrust. Spontaneous curvature or bending due to thermal or active fluctuations[9,23] are smaller than the structures we investigate here.

## Filament crossings & three-dimensional structure

High-speed confocal microscopy (Fig. 3a) reveals that the filaments in the ring are not confined to a single plane but overlap, forming an entangled structure typically 3–5 filaments high but several tens of filaments wide. In order to allow for overlap between filaments in the

simulations, we use a soft harmonic potential in a 2D framework (Fig. 3(b)). Repulsion sets in when the distance $r$ between two beads from distinct filaments is less than a threshold $d$. This ensures avoidance but allows crossings at some finite energy cost. Strictly 2D-confined experiments are challenging because physical confinement introduces boundary interactions that modify the filament behavior. Thus, by tuning the overlap parameter $d$, we numerically explore the effective shift from a scenario where there is no overlap to a situation where overlaps easily occur (see "Methods").

Figure 3c shows radial density profiles with increasing values of $d$, revealing a significant reduction in the peak density near the circular boundary. Plotting the difference between the densities at the peak and the center region as a function of $d$ (Fig. 3d), it becomes evident that there exists an optimal range in $d$ that leads to a well-defined and distinct ring formation. We rationalize the existence of this optimal value by noting that, for small $d$, filaments barely interact upon crossing, thus preventing aligning interactions. Conversely, increasing the value of $d$ beyond this optimal range reduces overlap and imposes a stricter two-dimensional confinement. In this scenario, filaments spontaneously arrange in traveling clusters analogous to those observed in two-dimensional glider suspensions[46–48] as well as simulations of self-propelled hard rods[49]. The chaotic dynamics of these clusters impair accumulation at the illumination boundary (Supplementary Movie 4). We quantify this feature in Fig. 3e, which reveals that the width of the ring grows linearly with $d$. As a result, time-averaged density profiles become more uniform in the illuminated region, while the residual density peak at the illumination edge can be traced back to intermittent trapping of the clusters at this location.

## Shapes with inhomogeneous boundary curvature

To uncouple size-dependent geometric factors from the effect of curvature, we performed experiments and simulations with elliptic, cornered and non-convex shapes (Fig. 4). Experiments and simulations in elliptic confinement with weak eccentricity yield ring-like patterns, with filaments accumulating at and aligning with the illumination boundary (Fig. 4b). Consistent with the picture of filaments behaving as secants of illuminated disks (Fig. 2d), an enlarged gap between the ring and light boundary is observed near the strongly curved apices of the ellipse. These characteristics also remain robust for non-convex shapes so long as they present sufficiently low curvature, as illustrated in Fig. 4h for a round corner concave square. In contrast, the presence of regions with curvature exceeding the threshold for ring formation (Fig. 2d) results in a qualitative change of the filament organization. For example, for an ellipse with strong eccentricity such as the one shown in Fig. 4c, the filaments no longer bend to follow the interface but rather splice into it, thus locally forming a splay deformation pattern (see Fig. 4a for a sketch of the deformation patterns and Appendix Fig. S6 for local orientation of the experimental snapshots). Remarkably, this transition from bend to splay is also observed in numerical simulations, so that it is purely induced by the change in curvature and does not rely on additional features like adhesion between the bacteria.

For shapes with straight edges and convex corners, as in regular polygons, curvature is focused to singular points. Also for these cases (Fig. 4d–g), accumulation at the boundaries is systematically observed. In addition, the corner angle determines the local nematic order pattern formed by the accumulated filaments. For obtuse angles down to right angles (see the trapezoid and square in Fig. 4d, e), the filament bundles bend to follow the edge, leaving a slight gap right in the corner. For acute angles (see the trapezoid, triangle and teardrop in Fig. 4e–g), however, filaments splice into the corner. The resulting splay deformation pattern then leads to a sudden reversal of the direction of rotation of the nematic order along the mask edge (Fig. 4a). In both simulations and experiments, the corner angle at which the filament deformation transitions from bend to splay is found to be about 90°.

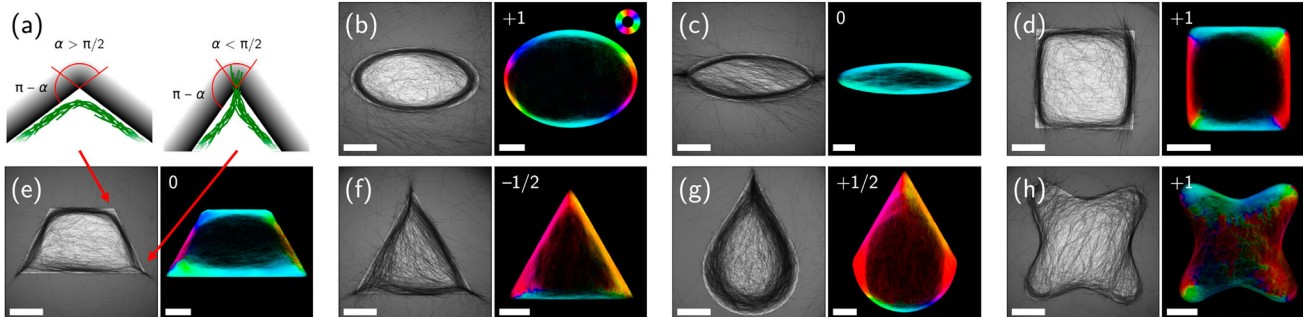

**Fig. 4 | Accumulation patterns for non-circular illumination. a** Schematic of filaments that bend or splice into corners with obtuse or acute angles, respectively. The red arrows point towards the experimental observations of the trapezoid in (**e**). **b**–**h** Results from experiments and simulations. The left panel shows stationary configurations experimentally obtained after ~17–21 h of illumination (scale bar is 1 mm). The right panel corresponds to the orientation of the time-averaged nematic order in numerical simulations of the active filament model (scale bar is 20 $\sigma$). The saturation has been adapted to reflect the time-averaged density so that dilute regions appear darker. The color code for the orientation is given in (**b**), while the winding number corresponding to each shape (see main text) appears at the top left corner.

To quantitatively relate the geometry of the virtual confinement to the morphology of the bacterial pattern, we consider the winding number associated with the circulation of the filament orientation along the boundary of the illuminated region. Since a bend (splay) deformation corresponds to a rotation of the filament pattern by an angle $\pi - \alpha$ ($-\alpha$) at a vertex with opening angle $\alpha$ (Fig. 4a), simple geometric arguments predict that the total rotation along a closed curve defining the boundary is given by $2\pi w$, where the winding number is

$$w = 1 - \frac{n_a}{2}. \tag{2}$$

Here, $n_a$ stands for the number of acute angles in the polygon (details in "Methods"). As shown in Fig. 4, Equation (2) is verified for all shapes previously described, so that the structure of the colony is fully determined by the geometry of the illumination pattern. Namely, in the absence of acute angles $w = 1$ (Figs. 1c, d and 4b, d), while for the flattened ellipse of Fig. 4c the two apices with high local curvature behave similarly to acute angles, leading to $w = 0$. A zero winding number is also obtained for trapezoids (Fig. 4e), while equilateral triangles and teardrop-like shapes give rise to $w = -\frac{1}{2}$ and $w = +\frac{1}{2}$, respectively (Fig. 4f, g). Note that, as convex polygons satisfy $n_a \leq 3$, the shapes displayed in Fig. 4 cover all possible values of $w$.

The above features are reminiscent of two-dimensional nematic liquid crystals, for which similar topological transitions between different frustrated defect patterns are induced by geometrical confinement[50]. Here, however, the bacteria are not subject to any physical boundary, but the mechanical confinement is self-generated by their emergent accumulation at the edges of the pattern. This effect only breaks down when the edge length becomes comparable to or smaller than the typical filament length, similar to the case of disk-shaped regions. Appendix Figs. S7 and S8 show densities and snapshots from simulations and experiments with smaller squares and triangles, respectively.

## Discussion

Our results demonstrate that filamentous cyanobacteria, which reverse their gliding direction in response to light gradients, exhibit emergent alignment and aggregation at the boundaries of convex light patterns. This behavior is robust against variations of curvature: corners are met by bending or splicing into them, depending whether they are obtuse or acute, respectively. Only when the overall size of the pattern becomes comparable to the filament length, boundary accumulation vanishes. Importantly, the light pattern only triggers pure direction reversals but does not re-orient the filament's body. This contrasts the case of steric interactions with physical walls, which generate aligning torques that are known to cause, e.g., the accumulation of trajectories of micro-swimmers like *Chlamydomonas reinhartii* and catalytic Janus particles along hard walls[39,51,52]. In our case, however, the confinement is non-mechanical and does not induce any aligning torques. Nonetheless, filaments globally accumulate at, and align along, edges of illuminated regions, which we understand as an emergent, collective effect.

We confirm the emergent and general character of this behavior with a minimal active filament model, for which three ingredients turn out to be essential: i. reversals at light gradients, ii. repulsion between filaments (leading to effective alignment), and iii. the ability of individuals to cross each other (absence of vertical confinement). If crossings of filaments are prevented, for example in a strictly 2D geometry, boundary accumulation and alignment are impeded by the formation of traveling clusters exhibiting chaotic dynamics in the illuminated domain.

Direction reversal may be considered one of the simplest possible responses of a micro-organism with one-dimensional motility. Reversing when sensing a declining viability of the environment is well known as a simple mechanism to accumulate in suitable conditions: scotophobic responses allow filamentous cyanobacteria to accumulate in lit regions, which can beautifully be visualized as "cyanographs"[28,53]. However, phototactic behavior requires re-orientation[54], which is lacking here. Our experiments with canonical convex light patterns demonstrate that scotophobic responses nonetheless trigger re-orientation and alignment, however as an emergent, collective phenomenon. Since boundary accumulation leads to higher local densities, it enables colonies to form structures more efficiently with limited amounts of filaments. Such aggregates are believed to shield colonies from external influences while enabling individuals to modulate their exposure by moving through the aggregate[17]. Thus our observation aligns with the properties of the freshwater benthic habitat of *O. lutea* where strong light gradients, e.g., from gravel or vegetation, are omnipresent.

Finally, we emphasize that, as the model does not include any specific detail about the biology of *O. lutea*, the collective effect we report here should be observable in a wide variety of systems, enabling filamentous organisms with purely one-dimensional motility of re-orientation, alignment, and aggregation according to sensory cues from their environment. From an engineering perspective, defining nematic textures by simple light stimuli ultimately provides a tool to create biomimetic mesh materials with programmable topological structures.

## Methods

### Cell cultivation

The original culture of *Oscillatoria lutea* (SAG 1459-3) was obtained from The Culture Collection of Algae at University of Göttingen and seeded in T175 culture flask with standard BG-11 (C3061-500ML, Merck) nutrition solution. The culture medium was exchanged for fresh BG-11 every 4 weeks. Cultures were kept in an incubator with an automated 12 h day (~10 µE, 18 °C) and 12 h night (dark, 14 °C) cycle, with a continuous 2 h transition. All experiments were performed at similar daytimes to ensure comparable phases in the circadian rhythm.

### Sample preparation

Roughly 1 mm³ of bacteria was seeded from the culture flasks into Petri dishes (627161, Greiner Bio-One) with diameter 35 mm and covered ~3 mL of fresh BG-11 solution. They were kept in the same incubator again for 24 h to allow for a homogeneous distribution on the bottom of the Petri dish and sealed right before the beginning of the experiment with parafilm.

### Imaging

Experiments were observed by a Nikon Ti2-E inverted microscope on a passive anti-vibration table, with transmitted brightfield LED illumination at about 25 µE intensity. Microscopy images were taken at 4× magnification (Plan Apo $\lambda$, NA = 0.2 and Plan Fluor, NA = 0.13) every 60 s for a duration of 18–24 h at a resolution of 4096 × 4096 pixels with a CMOS camera (Dalsa Genie Nano XL, pixel size 4.5 µm/px).

Between the light source and sample, a green filter (Lee filter No. 124) is placed with a circular punched hole of various sizes. The filter reduces the total light intensity by ~80% to 5 µE, and the remaining light is mainly in the green spectral range. Importantly, cyanobacteria do not significantly absorb in the green[38], so that the shaded regions appear effectively dark for the bacteria. An identical filter without any hole was placed in between the objective and the camera. This allows to resolve details in the dark part while still having enough light absorption for a patterned illumination. We performed extensive control experiments with gray filters and metal irises, finding no difference in filament behavior. The experiments with inhomogeneous boundary curvature (ellipse, square, triangle) were conducted by printing the illumination pattern on transparent film with a standard laser printer and no filter between the objective and the camera. The gray filter reduces the total light intensity by ~88% to 3 µE.

Confocal images were acquired by a Nikon Ti2-E inverted microscope with the bidirectional resonant scanner of an AX-R module, at a resolution of 2048 × 2048 pixels, using an S Fluor 40x Oil objective (NA = 1.3). For excitation of the autofluorescence, the 640 nm laser was used at the smallest possible intensity (0.1%). The pinhole size was ~1.0 Airy units, and the wavelength range of the detector was 663–738 nm.

### Density analysis

A pixel-wise maximum projection over time is performed to get a precise map of the illumination intensity, by which all frames are then normalized. This allows for a detection of density also in the region that was shaded by the green filter. Then, a morphological closing is used to reduce small dark artifacts from camera noise in the filaments. Completely black pixels (intensity $I = 0$) are set to $I = 1/255$ to prevent taking the logarithm of zero. The number of overlapping filaments $N$ for each pixel is then calculated according to *Beer-Lambert*

$$N = \frac{\log I}{\log I_1} \qquad (3)$$

where $I_1$ is the intensity of one filament and was determined in several calibration measurements to $I_1 \approx 0.7355$. Appendix Fig. S9 illustrates the procedure. At densities $N \gtrsim 8$, this signal starts to saturate because the image intensity drops below the camera sensitivity ($I_1^8 \approx 0.0856$).

Also, the density from defocused filaments appears blurred, which limits the resolution, especially for thicker rings where more and more filaments are located above the focal plane. For Fig. 2, density is then averaged azimuthally to obtain the average density vs. radius $r$, where midpoint and radius of the disk are calculated by fitting a circle to 4–8 points manually defined on the image of the disk.

To estimate the radius of the ring, the pixel-wise $N$ (see, e.g., Fig. 2a) is averaged over time, using only data from the second half of the experiment until the mask was removed, to exclude the initial transient phase. This signal is again azimuthally averaged. A Gaussian is fitted to the proximity of the peak density to avoid a possible bias from the different densities inside and outside of the ring. The range is determined by the half width at half maximum (HWHM) to the right side of the density profile, i.e., increasing $r$ and equally extending to the left side. The mean of the Gaussian fit is then taken as the ring radius $r_R$ and the error is estimated by the standard deviation of 3–8 independent repetitions of the measurements.

### Nematic order parameter

An overview for the process of the nematic order estimation is given in Fig. 5. To obtain a local orientation field, each frame is normalized by the maximum projection (see Fig. 5b), and the negative logarithm is taken. Then, a morphologically eroded (square kernel of size 9 × 9, about twice the filament diameter) version of the logarithmic image is subtracted from the image itself, to obtain local density variations (see Fig. 5c). Convolutions with dedicated kernels then return the mean density $\langle w \rangle$, the local offset $\langle x \rangle$ and $\langle y \rangle$ of the center of density, and the expectations $\langle x^2 \rangle$, $\langle y^2 \rangle$, and $\langle xy \rangle$ of the local distribution of density $N$ in Gaussian-weighted neighborhoods of each pixel. The local orientation is then calculated by

$$\Phi = \frac{1}{2} \arctan \left( \frac{-2 \left( \langle xy \rangle \langle w \rangle - \langle x \rangle \langle y \rangle \right)}{\langle x^2 \rangle \langle w \rangle - \langle x \rangle \langle x \rangle - \langle y^2 \rangle \langle w \rangle + \langle y \rangle \langle y \rangle} \right). \qquad (4)$$

Here, the convolutions $\langle \cdot \rangle$ are taken with Gaussian-weighted, normalized kernels of 13 × 13 pixels (standard deviation = 6 pixels) where $0.6 < I \leq 0.9$ (single filaments) or 401 × 401 pixels (standard deviation = 200 pixels), where $0.1 < I \leq 0.6$ (dense ring). For regions with $I < 0.1$ or $I > 0.9$, the mean orientation is not taken into account. This is realized by introducing two filters for the two different regions (single filaments and dense ring, see Fig. 5d, left), where the green color indicates the regions where the condition for $I$ is fulfilled. The convolutions are always taken for the full image shown in Fig. 5c and the filters are used to select the respective pixels. Both estimates are combined to yield the local orientation of the micrograph (Fig. 5d, right). The components of the nematic order tensor are obtained by

$$Q_{xx} = \langle \cos^2(\Phi) - 1/2 \rangle, \quad Q_{xy} = \langle \cos(\Phi) \sin(\Phi) \rangle \qquad (5)$$

and the norm of the nematic tensor is

$$Q = 2 \sqrt{Q_{xx}^2 + Q_{xy}^2}. \qquad (6)$$

Here, the local average $\langle \cdot \rangle$ is taken with a normalized square kernel over 201 × 201 pixels to ensure ensemble averages that contain several filaments. The components $Q_{xx}$ and $Q_{xy}$ are multiplied inside the average by the sum of the two filters to take only values into account that fulfill $0.1 < I \leq 0.9$. For the plots in Fig. 2, the nematic order is averaged over the azimuthal angle to obtain the mean nematic order vs. $r$.

### The active filament model

In our 2D simulation, we explore a system of $N_f$ self-propelled, semi-flexible filaments, each made from a chain of monomers. These monomers are connected by harmonic springs and are subject to a

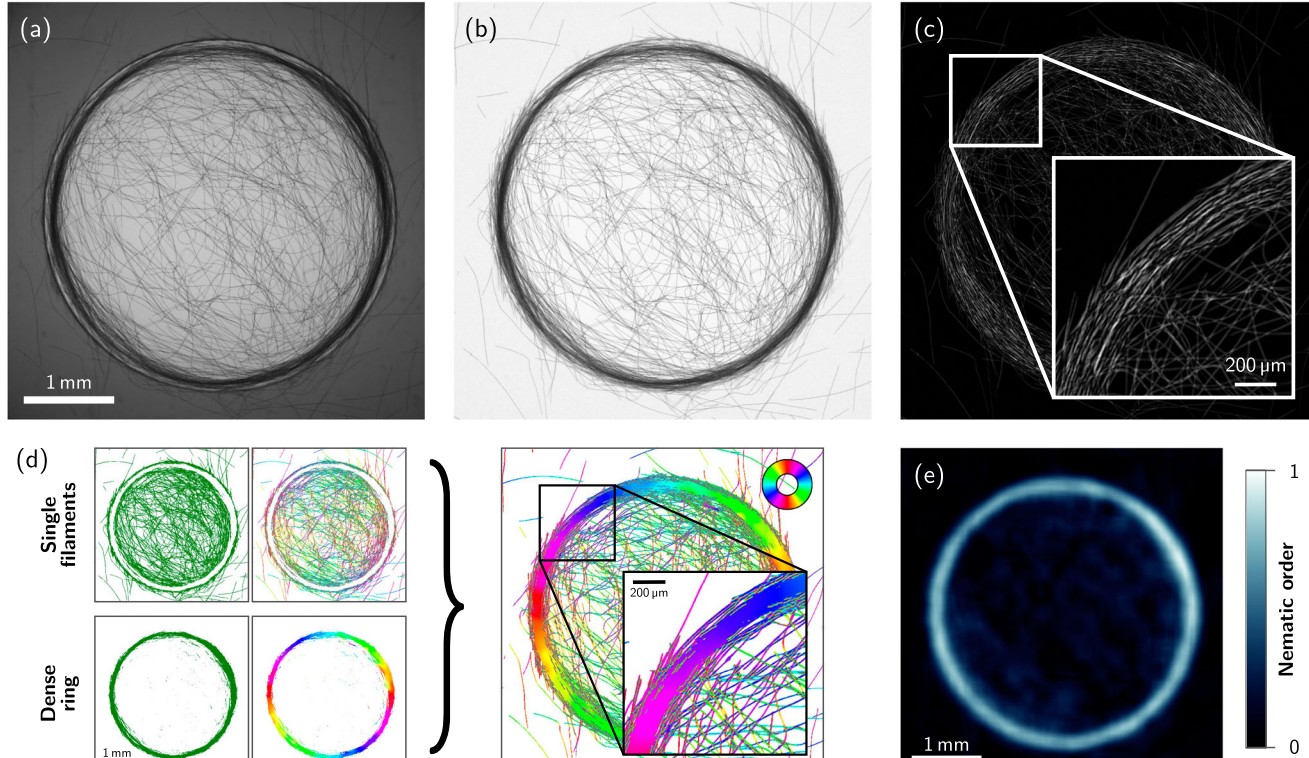

**Fig. 5 | Nematic order estimation for a dense ring aggregate. a** Raw micrograph of *O. lutea*, illuminated with a circular path of radius $R = 1.94$ mm after 10 h. **b** Raw micrograph divided by the maximum projection to compensate for varying intensities *I* across an image. **c** Local density variations obtained by subtracting a morphologically eroded (square kernel of size $9 \times 9$) version of a logarithmic image. **d** Local orientations in Gaussian-weighted neighborhoods. Depending on the intensity *I*, different filters (green color) are introduced for single filaments ($0.6 < I \le 0.9$) and inside the dense ring ($0.1 < I \le 0.6$). Combining both estimates yields the local orientation of the full micrograph, color indicates the orientation. **e** Norm of the nematic order tensor weighted by the sum of the two filters (single filaments + dense ring).

harmonic bending potential that determines the chain's flexibility. Due to the gliding nature of the motion of the bacteria, the model does not consider hydrodynamic interactions which are subdominant in this context.

In the overdamped regime, we describe the motion of each monomer's position $\mathbf{r}_i$ with the equation:

$$\zeta \dot{\mathbf{r}}_i = -\nabla_i U_s - \nabla_i U_b + \mathbf{F}_i^{\text{act}} - \nabla_i U_{\text{int}} + \mathbf{F}_i^{\text{rand}}. \tag{7}$$

The stretching potential, $U_s$, arises from the harmonic springs between monomers and is given by

$$U_s = \frac{\kappa_s}{2} \sum_{j=2}^{N} \left( r_{j,j-1} - \frac{\sigma}{2} \right)^2. \tag{8}$$

where $r_{j,j-1} = |\mathbf{r}_j - \mathbf{r}_{j-1}|$ is the distance between the *j*-th and (*j*−1)-th monomer belonging to the same chain. $\sigma/2$ represents the typical distance between two monomers within the same chain. The harmonic bending potential $U_b$ with stiffness $\kappa_b$ controls bond flexibility, and is defined as

$$U_b = \frac{\kappa_b}{2} \sum_{j=2}^{N-1} (\theta_j - \pi)^2, \tag{9}$$

where $\theta_j$ represents the angle formed by a consecutive triplet of monomers $(j-1, j, j+1)$, formally defined as

$$\theta_j = \cos^{-1} \left( \frac{\mathbf{r}_{j-1,j} \cdot \mathbf{r}_{j,j+1}}{r_{j-1,j} r_{j,j+1}} \right). \tag{10}$$

Self-propulsion is induced by the active force $\mathbf{F}_i^{\text{act}}$, given by

$$\mathbf{F}_i^{\text{act}} = f_{\text{act}} \hat{\mathbf{t}}_i, \tag{11}$$

where $f_{\text{act}}$ determines its magnitude, and

$$\hat{\mathbf{t}}_i = \frac{1}{2} \left( \frac{\mathbf{r}_{i+1,i}}{r_{i+1,i}} + \frac{\mathbf{r}_{i,i-1}}{r_{i,i-1}} \right) \tag{12}$$

is the unit vector tangent to the chain at the *i*-th monomer. To allow overlaps between filaments, we use a soft isotropic repulsion, such that the corresponding interaction potential is given by

$$U_{\text{int}}(r) = \begin{cases} U_0(r/d - 1)^2 & \text{if } r \le d, \\ 0 & \text{otherwise}. \end{cases} \tag{13}$$

Here, *r* represents the distance between two monomers belonging to different filaments, and the parameter *d* specifies the interaction range. The last contribution to the r.h.s. of Eq. (7), $\mathbf{F}_i^{\text{rand}}$, accounts for thermal fluctuations. Namely, $\mathbf{F}_i^{\text{rand}} = \sqrt{2D\zeta^2} \boldsymbol{\xi}_i$, where the $\boldsymbol{\xi}_i$ are uncorrelated zero-mean Gaussian white noises with unit variance, while *D* is the associated translational diffusivity.

The biased reversals of filaments at illuminated boundaries were implemented as follows: When the head of a given filament propelling outside of the illuminated region crosses the edge at time *t*, the direction of motion of all its monomers is reversed at time $t + dt$ ($f_{\text{act}} \rightarrow -f_{\text{act}}$), where d*t* denotes the time step used in simulations. On the other hand, filaments entering the illuminated boundary do not experience reversals. Note that this

**Table 1 | Simulation parameters corresponding to the data shown in the main text (unless specified otherwise)**

| Parameter | $\tilde{\kappa}_s$ | $\tilde{\kappa}_b$ | $\tilde{U}_0$ | $\tilde{d}$ | $R/\sigma$ | $\tilde{D}$ |
|---|---|---|---|---|---|---|
| Value | 500 | 5 | 2.4 | 0.2 | 60 | $5 \cdot 10^{-6}$ |

procedure does not induce any rotation of the filament body so that, up to thermal fluctuations, isolated filament undergoing reversals exactly follow their incoming trajectory in the opposite direction. Since filaments are regularly pushed out of the illuminated domain by other filaments, allowing single filaments to escape with a finite probability does not qualitatively influence the structure formation at boundaries.

All simulations were performed in a two-dimensional periodic box of size $188\,\sigma$. To account for the polydispersity of the filament lengths observed in the experiments (Appendix Fig. S1), in our simulations the filament lengths were drawn from a Poisson distribution with a fixed mean $20\,\sigma$. As the number of filaments was fixed to $N_f = 1020$, the simulated systems comprised on average $\approx 20{,}000$ monomers. Choosing $\sigma$ and $\sigma\zeta/f_{act}$ as units of length and time, respectively, the dimensionless Eq. (7) depends on five non-dimensional parameters, namely

$$\tilde{\kappa}_s = \frac{\kappa_s \sigma}{f_{act}}, \quad \tilde{\kappa}_b = \frac{\kappa_b}{\sigma f_{act}}, \quad \tilde{U}_0 = \frac{U_0}{\sigma f_{act}}, \quad \tilde{D} = \frac{k_B T}{\sigma f_{act}}, \quad \tilde{d} = \frac{d}{\sigma}.$$

The parameter values chosen in simulations are reported in Table 1. The equations of motion (7) were integrated with a time step $dt = 10^{-3}$ for typically $10^7$ simulation steps, which we checked was much larger than needed for the dynamics to reach a steady state. The large value of $\tilde{\kappa}_s = 500$ reported in Table 1 ensures marginal stretching of the filaments. We moreover checked that the results reported in the main text do not depend on any fine-tuning of the model parameters, as they remain qualitatively unchanged when, e.g., the filament stiffness $\kappa_b$ or the self-propulsion force $f_{act}$ is changed.

For sufficiently large $U_0$ and $d$, filaments experience strong repulsive forces that prevent overlaps and thus reproduce a strictly 2D confinement. Decreasing either of these two parameters, we observe crossing between filaments similar to those reported in the experiments (see Fig. 1d). Lowering $U_0$ at fixed $d = \sigma/2$ for the filament density used in our simulations, this transition quickly leads to a vanishing of the interactions between filaments, leading to structureless aggregates in the illuminated region. On the other hand, decreasing $d$ at fixed $U_0$ allows for a larger intermediate regime where filaments can cross each other while still effectively aligning nematically when their density reaches large enough values. As discussed in the main text and shown in Fig. 3, this regime where $0.15 \lesssim d/\sigma \lesssim 0.35$ precisely corresponds to the emergence of nontrivial patterns at the illumination boundary.

The data presented in Fig. 2d was obtained by varying the radius of the illuminated disk in the range $[5\,\sigma; 80\,\sigma]$ while keeping the system size fixed. In Fig. 4, we illustrate various geometries with the following specifications: ellipse with major and minor radii of $55\,\sigma$ and $40\,\sigma$, respectively (a), ellipse with major and minor radii of $66\,\sigma$ and $15\,\sigma$, respectively (b), square of side length $50\,\sigma$ (c), trapezoid with height $55\,\sigma$, length of the first base $124\,\sigma$, and of the second base $62\,\sigma$ (d), equilateral triangle with height $80\,\sigma$ (e), and a teardrop made from one equilateral triangle with height $80\,\sigma$ and a semicircle with radius $23\,\sigma$ (f).

### Nematic patterns in convex polygons

As described in the main text, boundary accumulation induced by convex light patterns is characterized by a bending (splicing) of the filaments at corners with obtuse (acute) angles. To evaluate the total circulation of the filament orientation along the edge of a polygon, we note that inside a corner with opening angle $\alpha$, the filament orientation

in the boundary structure may change either by $\pi - \alpha$ or by $-\alpha$, corresponding to bend and splay, respectively (see Fig. 4a). Considering a convex polygon with $n_a$ acute and $n_o$ obtuse angles, the winding number is thus given by

$$w = \frac{1}{2\pi}\left[\sum_{i=1}^{n_a}(-\alpha_i) + \sum_{j=1}^{n_o}(\pi - \alpha_j)\right].$$

Noting that the angles of the polygon satisfy $\sum_{i=1}^{n_a}\alpha_i + \sum_{j=1}^{n_o}\alpha_j = (n_a + n_o - 2)\pi$, it is then straightforward to recover Eq. (2) of the main text.

### Data availability

Raw experimental data generated in this study is provided in a data repository at https://doi.org/10.17617/3.NTKSHK. Source data of the figures are provided in the Source Data file. Source data are provided with this paper.

### Code availability

We made the image analysis code that was used for computing the density and nematic order from the experimental images publicly available. We additionally publish a minimal working script for the numerical simulations. The code is available at https://doi.org/10.17617/3.NTKSHK.

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

## Acknowledgements

The authors gratefully acknowledge Maike Lorenz and the Algae Culture Collection (SAG) in Göttingen, Germany, for providing the cyanobacteria species *O. lutea* (SAG 1459-3) and technical support. We also thank D. Strüver, M. Benderoth, W. Keiderling, and K. Hantke for technical assistance, culture maintenance, and discussions. We gratefully acknowledge discussions with M. Prakash, A. Vilfan, K. R. Prathyusha, and G. Fava. We acknowledge support from the Max Planck School Matter to Life and the MaxSynBio Consortium which are jointly funded by the Federal Ministry of Education and Research (BMBF) of Germany and the Max Planck Society. S.K. acknowledges funding from the German Research Foundation (DFG, Project No. 519479626).

## Author contributions

M.K. and S.K. designed the experiments, M. K. and F. P. performed the experiments and analysed data, L.A., P.B., R.G., and B.M. designed the simulations. L.A. wrote the simulation code, ran the model, and analysed output data. S.K. and B.M. conceived the study, P.B. and R.G. provided conceptual advice. All authors contributed to the interpretation of the data, discussions of the results and writing of the manuscript.

## Funding

## Competing interests

The authors declare no competing interests.

## Additional information

**Supplementary information** The online version contains
supplementary material available at

Benoît Mahault or Stefan Karpitschka.

**Peer review information** *Nature Communications* thanks Nitesh Arora,
Saad Bhamla and the other, anonymous, reviewers for their contribution
to the peer review of this work. A peer review file is available.

