## [Peer Review File · Nature Communications]

Collective self-caging of active filaments in virtual confinementREVIEWER COMMENTS

Reviewer #1 (Remarks to the Author):

Kurjahn, Abbaspour, et al. experimented with cyanobacterium *O. lutea* and showed that these elongated microorganisms actively self-caged at light boundaries. In the experiments, light gradients are created using bright field illumination and masking, which creates sharp gradients at the dark/light boundary. The boundary acts as a virtual confinement because individual filaments are scotophobic, i.e., they reverse upon encountering variations in illuminance. They rationalized their findings using an active polymer model of filaments that reverse at light gradients and exhibit steric interactions but can cross each other.

The paper represents a timely contribution to a very active area of research at the intersection of active matter and microbiology: the study of pattern formation by elongated organisms. The paper contributes a new, interesting idea of virtual confinement and convincingly identifies minimal ingredients needed to reproduce the observed patterns. I recommend publication in Nature Communications.

I only have minor comments:

1. The introduction raises the expectation that the authors study the flexibility of filaments in the paper. I specifically mean the gap sentence: "...little is known about how they are affected by the strong shape anisotropy and flexibility of filamentous agents." However, the flexular rigidity was varied in this work. I suggest the authors clarify the scope of the paper to avoid confusion.

2. The authors should provide more information about the organisms' lifestyles. Do they typically live on surfaces or are suspended in water?

3. In the experiments, do the filaments always sink to the bottom? Are they ever positively buoyant?

4. Fig. 3(b) label. It took me a while to realize that the 'r' label on the x-axis is the name of the axis variable, not a specific distance $r > d$. I suggest using the same style as in other panels.

5. In the conclusions, I would stress that by the 'ability to cross each other', the authors have in mind the three-dimensional effect of filaments moving above or below each other.

6. Beer-Lambert law. Can the authors explain how the individual filament intensity was calibrated in more detail? How obvious is it that Beer-Lambert law holds for filamentous bacteria? I agree that the intensity I is a monotonic function of filament density, but it is unclear to me whether N can be estimated using Eq. (3).

7. Given that Nature Communications is a multidisciplinary journal, I suggest the authors try to better connect their findings with the lifestyle of *O. lutea*. Under what environmental conditions or life stages do the authors think the observed self-caging behavior may be relevant from an ecological or evolutionary perspective?

8. Code availability. Can the authors make the image analysis code used to compute the density and nematic order publicly available?

Reviewer #4 (Remarks to the Author):

The paper by Kurjahn et al. shows an interesting light-controlled emergent phenomena in gliding cyanobacteria, which they supplement with a mathematical model. The main discovery here is that these 'light-reversing' active filaments aggregate at the boundary – allowing the team to use the light to 'sculpt' these filaments at the edge of a 'designer' shape.

Overall this is an interesting that builds on emerging interest in active filaments by introducing the novelty of using light to aggregate these 'agents' at boundaries. The technical aspects are tightly knit and solid. For example, In Fig 2, the correlation of the peak density radius with the secant close to the filament length is a good observation and the authors also show good agreement of experiments with the modeling.

However, we may have some concerns that may require a revision to really make a case for why this paper is suitable for a multidisciplinary journal like Nature Comm

1. It seems that the paper is hurried up. The authors could have added more interesting results and could have shown some potential applications. Essentially the authors have some early results, but these simulations could have been further expanded into charting some new ideas. For example, could these be used to establish order in a 3d droplet? Or a 3d toroid? In nature these cyanobacteria indeed make large balls or tangled living mats – and the authors could easily use their simulations to explore this phenomena.
2. The paper reads more like a PRL paper – focused on a physics audience. The broader implications, utility of their findings and **biological relevance** are lacking. For example the authors do discuss some CO2 capture by these algae or connection to aggregation in sunlight (cyanographs) – but no connection or even hint to understanding a broader appreciation of these extraordinary living systems is offered – besides a physicist, why should anyone else care? (being dramatic here since this is Nat Comm and not PRL)
3. The same goes for the introduction and motivation for the work – it is weakly written. In the Intro, this sentence tries to discuss the gap in the literature, "*While the statistical physics principles behind these emergent behaviors are well understood for simple, compact agents [31, 32], little is known about how they are affected by the strong shape anisotropy and flexibility of filamentous agents.*" For example these ideas could be important for soft flexible robotics or roots etc – and a broader vision could really help put their work in a larger context.

Minor Comments:

1. Is there any effect of light on the motion of cyanobacteria, it's not considered in the modeling.
2. How will the shapes with inhomogeneous boundary curvature in Fig 4 look for different sizes?
3. Please give a step-by-step demonstration of nematic order estimation from images.

4. What is the finite probability of cyanobacteria moving into the dark region from the illuminated region used in the simulations? And how it is related to the experiment? Do they have similar probabilistic behavior in experiments?
5. It would be interesting to look at how the cyanobacteria will aggregate when there are multiple illuminated regions in the dish.
6. The authors have shown the dissolution of cyanobacteria when the mask is removed. How does the cyanobacteria assembly evolve when the mask is shifted to create the illuminated region at a new position?
7. What happens when the shapes are non-convex?
8. Can there be some application of these findings? Or can it be connected to some assemblies already seen in nature?
9. Will the overlapping of the cyanobacteria change when we have more of them in the petri dish? How will their aggregation change?
10. How is the self-propulsion force chosen in simulations? Does it change with time? How does it correlate with the experiments?
11. The word "scotophobic" is defined on the last page – this should be up front and center – this is not a common term and is central for the reader to appreciate these findings – burying it in the conclusion does not help the author's case.
12. Why the winding number (eq 2) is discussed in detail is not very clear – is it really important for the paper? What is the point of it?

Collective self-caging of active filaments in virtual confinement: Report

In this article by Kurjahn et al, the authors pair experiments and simulations to look at 'scotophobic' i.e. darkness-avoiding filamentous cyanobacteria. These long, thin filaments reverse gliding direction when coming into contact with darkness. The authors take advantage of this by constructing round and polygonally shaped patterned regions of illumination. The filaments then accumulate tangentially to the boundaries, forming a ring for circular regions and patterns that include discontinuities for polygonal shapes.

The authors provide a convincing semi-quantitative explanation of the filament orientation and density patterns by pointing out that filament motion patterns will cause them to accumulate along secants to the circle.

They then pair experiment with a qualitative active polymer simulation to compute nematic orientation, density profiles and the patterns for systems with sharp corners. For the latter, they propose a counting rule for the winding number, $w = 1 - \frac{n_a}{2}$ that counts the number of acute angles n_a where the filaments have what looks like a splay defect to me.

The accompanying simulation is qualitative, with no real attempt at matching the experiment quantitatively:

- Why is this not in fully reduced units? The choice of units is related to a thermal monomer diffusion time which looks irrelevant here. The natural choice would be using a unit friction coefficient, particle size and then one more number to set a force scale, e.g. the active force magnitude. Instead, there are what seems to be a bunch of arbitrary numbers with no attempt at understanding scales.
- I would have expected the relevant scaling for the secant density profiles to be the persistence length ξ of the filament scaled by the radius R of the circle, or if very stiff, its length (from the movie, I can just about see that the filament is nearly but not totally stiff). There is no discussion about any of this.
- The relatively weak interactions between filaments are modelled here by a soft harmonic potential with range d and strength U_0 . This accounts for filaments that can overlap. There is a transition towards filament alignment with d here, but U_0 is a fixed number. I would have kept d fixed and then expected that $U_0 d / f_{act}$ would act as a scale of the alignment strength.

For the non-circular confined systems: How can the sharp corner and winding number calculation be linked to the usual language of topological defects? All of the systems shown here are isomorphic to a circle with Euler number 1. Then with a tangent boundary condition, I would expect the sum of the charges of the topological defects in the system to always be 1 as well. Of course, the corners may introduce issues here, and giving that filaments can cross local nematic order may be hard to define everywhere. I tried to construct the following defect patterns, but it is not quite satisfactory:

Finally, the structure of the article is a bit strange. There is a long paragraph about the simulations in the initial 'Patterned illumination' section, starting with '*To confirm that the formation of the ring*'. This feels out of place,

One more minor point: Please define 'scotophobic' the first time you use it

All in all this is a nice experiment, but I am not sure if in its current state the level of model and explanation is quite sufficient for a high profile publication.

Reply to Reviewer #1

Reviewer 1: *Kurjahn, Abbaspour, et al. experimented with cyanobacterium *O. lutea* and showed that these elongated microorganisms actively self-caged at light boundaries. In the experiments, light gradients are created using bright field illumination and masking, which creates sharp gradients at the dark/light boundary. The boundary acts as a virtual confinement because individual filaments are scotophobic, i.e., they reverse upon encountering variations in illuminance. They rationalized their findings using an active polymer model of filaments that reverse at light gradients and exhibit steric interactions but can cross each other.*

The paper represents a timely contribution to a very active area of research at the intersection of active matter and microbiology: the study of pattern formation by elongated organisms. The paper contributes a new, interesting idea of virtual confinement and convincingly identifies minimal ingredients needed to reproduce the observed patterns. I recommend publication in Nature Communications.

Reply: We are grateful for the reviewer’s positive evaluation of our work and the recommendation to publish our work. We have revised our manuscript according to the comments as detailed below.

Reviewer 1: *I only have minor comments:*

1. *The introduction raises the expectation that the authors study the flexibility of filaments in the paper. I specifically mean the gap sentence: “..little is known about how they are affected by the strong shape anisotropy and flexibility of filamentous agents.” However, the flexular rigidity was varied in this work. I suggest the authors clarify the scope of the paper to avoid confusion.*

Reply: We have rephrased this sentence in the introduction:

...In contrast to compact photoresponsive agents like *E. coli* [30, 31] or self-thermophoretic colloids [32], for which light-controlled aggregation is well studied [33, 34], filamentous agents generally entail additional emergent phenomena that are currently not well understood [10, 35].

Reviewer 1: 2. *The authors should provide more information about the organisms’*

lifestyles. Do they typically live on surfaces or are suspended in water?

Reply: We thank the referee for asking this question, which should indeed be answered in the manuscript. The following sentences were added to the description of the experiments (page 2):

The response of the filamentous cyanobacterium *Oscillatoria lutea* to light gradients was observed with the experimental setup sketched in Fig. 1(a). *O. lutea* is a tychoplanktic freshwater species that forms sessile and floating colonies [36]. Individual filaments are negatively buoyant, but aggregates may develop gas bubbles, enclosed in a mesh of filaments and secreted slime, that cause flotation. The mechanical properties and collective behavior of *O. lutea* have recently been investigated [9, 23, 24, 37]. Filaments are about 5 μm wide, flexible, and exhibit bi-directional gliding motility that is sustained in complete darkness. Filaments were contained ...

Reviewer 1: 3. *In the experiments, do the filaments always sink to the bottom? Are they ever positively buoyant?*

Reply: This is indeed a very interesting question. We never observed individual filaments to be positively buoyant in our experimental conditions, although buoyancy regulation through gas vacuoles has been discussed for similar species in the literature. Aggregates of many filaments, though, typically develop gas bubbles due to photosynthetic activity, embedded in a mesh of filaments surrounded by polysaccharide slime. Such structures develop positive buoyancy and may eventually detach from submersed surfaces, giving the species their tychoplanktic character.

We added a comment to the description of the aggregation experiments (page 2, same paragraph as for the last question).

Reviewer 1: 4. *Fig. 3(b) label. It took me a while to realize that the ‘r’ label on the x-axis is the name of the axis variable, not a specific distance $r > d$. I suggest using the same style as in other panels.*

Reply: We thank the referee for pointing this out. We changed the label in panel (b) of Fig. 3 accordingly (see below).

Reviewer 1: 5. *In the conclusions, I would stress that by the ‘ability to cross each other’, the authors have in mind the three-dimensional effect of filaments moving above or*

Fig. R1: Updated **Fig. 3** of the main manuscript with new label on the x -axis of panel (b).

below each other.

Reply: We thank the referee for this suggestion. Indeed, although in dilute conditions filaments are bound to the bottom surface of the Petri dish, the patterns that we observe are not strictly two-dimensional, but consist of several layers of filaments as shown in Fig. 3(a) of the main text. Following the referee’s suggestion, this part of the discussion was modified as:

We confirm the emergent and general character of this behavior with a minimal active filament model, for which three ingredients turn out to be essential: i. reversals at light gradients, ii. repulsion between filaments (leading to effective alignment), and iii. the ability of individuals to cross each other (absence of vertical confinement). If crossings of filaments are prevented, for example in a strictly 2D geometry, boundary accumulation and alignment are impeded by the formation of traveling clusters exhibiting chaotic dynamics in the illuminated domain.

Reviewer 1: 6. Beer-Lambert law. Can the authors explain how the individual filament intensity was calibrated in more detail? How obvious is it that Beer-Lambert law holds for filamentous bacteria? I agree that the intensity I is a monotonic function of filament density, but it is unclear to me whether N can be estimated using Eq. (3).

Reply: This is indeed a very relevant technical question. Light absorption is an established standard to quantify biomass in bioreactors or small-scale cultures according to the Beer-Lambert law. Filamentous cyanobacteria are strongly pigmented, so absorption is strong compared to scattering. In our case, we calculate the typical absorption of an individual filament by comparing the image intensity \mathcal{I}_1 for a single filament to the reference intensity \mathcal{I}_0 at the same location but without a filament. Importantly, the filament diameter is always larger than the optical resolution or the size of a single pixel. For computational efficiency, we first re-scale the image by the reference intensity to obtain $I = \mathcal{I}/\mathcal{I}_0$ and the corresponding single-filament intensity $I_1 = \mathcal{I}_1/\mathcal{I}_0$, to compensate for spatially inhomogeneous illumination. The number of overlapping filaments N is then estimated according to $N \approx \log I / \log I_1$, which is typically well resolved (Fig. R2 b). The modified gamma-correction of sRGB color-space cancels to a very good approximation because the offset is negligible.

Fig. R2: Local filament number density per pixel. (a) Zoom into a micrograph of *O. lutea*, scale bar is 100 μm . (b) Number of overlapping filaments N per pixel calculated with Eq. (3) and rounded to the next integer.

There are of course limitations to this technique. As can be seen in Fig. R2, defocused filaments are not detected in the discretized signal because the absorption signal is spread out over the defocusing disk. However, the integral absorption remains constant, so this effect is largely compensated for when integrating the density signal. Thus, we are confident

that N accurately describes the filament density as long as densities (vertical stacking of filaments) remains small enough so that defocusing remains below the integration area. Yet, sharp density steps from defocused regions would appear blurred, of course. Additionally, each filament exhibits a texture in the image, resulting from pigmentation variation, scattering on internal structures, or diffraction on the cells themselves. Therefore, the signal is of course only an estimate of the mean density, with limited spatial resolution. Finally, for high densities, the camera signal saturates to some small black value, and higher densities can no longer be resolved. Overall, we observed the onset of saturation effects once the density exceeded ~ 8 filaments, well below the range of data shown in Fig. 2 of the main manuscript.

We have added a paragraph stating the limitations of the technique to the methods section:

... where I_1 is the intensity of one filament and was determined in several calibration measurements to $I_1 \approx 0.7355$. At densities $N \gtrsim 8$, this signal starts to saturate because the image intensity drops below the camera sensitivity ($I_1^8 \approx 0.0856$). Also, the density from defocused filaments appears blurred, which limits the resolution especially for thicker rings where more and more filaments are located above the focal plane. For Fig. 2, density is then averaged
...

Reviewer 1: 7. *Given that Nature Communications is a multidisciplinary journal, I suggest the authors try to better connect their findings with the lifestyle of *O. lutea*. Under what environmental conditions or life stages do the authors think the observed self-caging behavior may be relevant from an ecological or evolutionary perspective?*

Reply: This question aligns with the main critique of referee 2, and we thank the referee for giving us the opportunity to make our work more appealing to a broad readership. Nonetheless we would like to point out that it is not only the ecological or biotechnological relevance of our model organism that makes the work interesting for a general audience. Rather, the interdisciplinarity arises from the generality of the phenomenon: Alignment relative to external cues emerges for any elongated organism with responsive, bi-directional motility, which encircles a huge portion of the microbial domain.

O. lutea serves as a non-toxic model system for bloom forming benthic species [29] which usually dwell in or on soil submersed in fresh water, frequently shaded by stones or other vegetation, so naturally exposed to light patterns. During blooms, aggregates enclosing

gas bubbles from photosynthetic byproducts become buoyant, detach from the benthos and rise to the surface of the water body [21]. In general, aggregation is assumed to be a decisive step in the growth of colonies of filamentous cyanobacteria, but the aggregation mechanisms of such tychoplanktic species have not been investigated, apart from a very recent publication by Faluweki et al, which we cite in our manuscript. However, the effect of light patterns were not investigated in that study.

It is commonly assumed that life in an aggregate offers an evolutionary advantage, e.g. for shared EPS secretion or light exposure regulation by self-shading: a compact aggregate exposes less surface to direct illumination and allows individuals to modulate their exposure by migrating within the aggregate [17]. But also detaching from a surface and migrating vertically through a water body can only be achieved collectively by these organisms [21]. Such migration is essential for dispersal and exploration of new habitats for which surface-bound gliding at a few micrometers per second is not efficient.

Light patterns on the scale we investigated here are naturally ubiquitous in typical habitats of filamentous cyanobacteria. Aggregating on a light boundary greatly enhances the chance of encountering peers or existing aggregates as compared to a random exploration of a two-dimensional space because individual filaments are not capable of active steering. Using naturally emergent mechanisms is therefore the only way to enhance aggregation.

In order to better reflect the broad relevance of our findings, we have modified the introduction and the discussion.

Introduction (page 1):

Emergent collective behaviors of motile biofilaments are ubiquitous across scales in nature, from cytoskeleton constituents [1–4] to colonies of elongated bacteria [5–10] and even millimeter-sized worms [11, 12], with promising applications in synthetic, bio-inspired materials [13–15]. In this context, filamentous cyanobacteria are currently attracting significant attention [9, 10, 16]. These microorganisms are indeed among the oldest, yet still most abundant phototrophic prokaryotes on Earth, fixing vast amounts of atmospheric carbon by photosynthesis [17, 18]. They comprise a wide variety of phenotypes, habitats, and lifestyles, covering benthic and planctonic forms in fresh and salt water, as well as adaptations to extreme conditions like hypersaline lakes or hot springs [17]. Driven by climate change and eutrophication, harmful cyanobacterial blooms are intensifying in recent years [19, 20]. Such blooms typically develop when submersed populations grow to a critical density, upon which individual filaments aggregate and eventually rise to the surface of the water body [10, 17, 21]. The role of tychoplanktic species, i.e. organisms that usually

dwell on submersed surfaces but eventually aggregate, detach, and rise to the water surface, remains to be elucidated [17]. Beyond their major ecological impact [17, 22], cyanobacteria are gaining relevance as renewable energy source from photobioreactors or for biochemicals production [18].

Filamentous cyanobacteria are long and flexible [23, 24], containing up to several hundred, linearly stacked cells. Many species glide actively along solid surfaces or each other [10], exhibiting photo- and scotophobic responses [25] i.e., reversals of their direction of motion when they experience an increase or decrease in illuminance, respectively. These phenomena are currently not understood in detail [24, 26, 27], but appear to be crucial for aggregation, which, besides its role in blooming, is believed to provide light exposure regulation [28] and protection from other environmental factors [17]. Yet, the genesis of aggregation remains largely unknown [10], especially for benthic and tychoplant species that are usually subject to spatially inhomogeneous illumination [9, 29]. In contrast to compact photoresponsive agents like *E. coli* [30, 31] or self-thermophoretic colloids [32], for which light-controlled aggregation is well studied [33, 34], filamentous agents generally entail additional emergent phenomena that are currently not well understood [10, 35].

Discussion (page 6):

... Since boundary accumulation leads to higher local densities, it enables colonies to form structures more efficiently with limited amounts of filaments. Such aggregates are believed to shield colonies from external influences while enabling individuals to modulate their exposure by moving through the aggregate [17]. Thus our observation aligns with the properties of the freshwater benthic habitat of *O. lutea* where strong light gradients e.g. from gravel or vegetation, are omnipresent.

Finally, we emphasize that, as the model does not include any specific detail about the biology of *O. lutea*, the collective effect we report here should be observable in a wide variety of systems, enabling filamentous organisms with purely one-dimensional motility of re-orientation, alignment, and aggregation according to sensory cues from their environment. From an engineering perspective, defining nematic textures by simple light stimuli ultimately provides a tool to create biomimetic mesh materials with programmable topological structure.

Reviewer 1: 8. Code availability. Can the authors make the image analysis code used

to compute the density and nematic order publicly available?

Reply: We added a code availability section to the manuscript (page 10) and made the image analysis code publicly available. The public link in the revised manuscript will be activated upon publication of the manuscript. In case the reviewer would like to inspect the code in advance, we share here a draft link to the repository: <https://edmond.mpg.de/privateurl.xhtml?token=943a17c5-042e-4c9e-974d-b8cfe736f224>

Code availability. We made the image analysis code that was used for computing the density and nematic order from the experimental images publicly available. We additionally publish a minimal working script for the numerical simulations. The code is available at <https://doi.org/10.17617/3.NTKSHK>.

Reply to Reviewer #2

Reviewer 2: *The paper by Kurjahn et al. shows an interesting light-controlled emergent phenomena in gliding cyanobacteria, which they supplement with a mathematical model. The main discovery here is that these ‘light-reversing’ active filaments aggregate at the boundary – allowing the team to use the light to ‘sculpt’ these filaments at the edge of a ‘designer’ shape.*

Overall this is an interesting that builds on emerging interest in active filaments by introducing the novelty of using light to aggregate these ‘agents’ at boundaries. The technical aspects are tightly knit and solid. For example, in Fig. 2, the correlation of the peak density radius with the secant close to the filament length is a good observation and the authors also show good agreement of experiments with the modeling.

Reply: We thank the referee for this positive evaluation of our work. Below we answer each point of criticism in detail.

Reviewer 2: *However, we may have some concerns that may require a revision to really make a case for why this paper is suitable for a multidisciplinary journal like Nature Comm.*

1. It seems that the paper is hurried up. The authors could have added more interesting results and could have shown some potential applications. Essentially the authors have some early results, but these simulations could have been further expanded into charting some new ideas. For example, could these be used to establish order in a 3d droplet? Or a 3d toroid? In nature these cyanobacteria indeed make large balls or tangled living mats – and the authors could easily use their simulations to explore this phenomena.

Reply: We thank the referee for suggesting the addition of potential applications and charting some new ideas. We have re-written the introduction and discussion to include these aspects. Below, the Referee specifically asked for applications, where we replied in detail. In short, our work shows how a nematic texture can be generated in an engineered living mesh material, leading to a programmable topology. Another possible application would be density and migration control in cell culture experiments, because consecutive bright and dark regions, in combination with boundary alignment, effectively build up a barrier against filaments entering or leaving specific regions. In terms of ecology, *O. lutea* is

a benthic to tychoplanktic freshwater species, naturally experiencing strong light gradients in its natural habitat. Thus, the mechanism we describe may facilitate aggregation, which in turn is believed a protection mechanism from environmental factors [17].

In terms of 3D droplets or toroids, the reviewer is certainly proposing interesting extensions for our work. However, we note that our simulation framework is purely two-dimensional, while three-dimensional effects (like filament crossings) are implemented in an effective way. The simplicity of our model is intentional because we do not aim at reproducing real aggregates in silico. Instead, the main contribution of our minimal numerical model is to highlight the generality and the emergent nature of the boundary accumulation phenomenon observed in the experiments, which would be concealed in an overly complex model. Therefore, simulating fully three dimensional structures like droplets, although an interesting avenue to explore in the future, is outside of the scope of this work.

Reviewer 2: *2. The paper reads more like a PRL paper – focused on a physics audience. The broader implications, utility of their findings and **biological relevance** are lacking. For example the authors do discuss some CO₂ capture by these algae or connection to aggregation in sunlight (cyanographs) – but no connection or even hint to understanding a broader appreciation of these extraordinary living systems is offered – besides a physicist, why should anyone else care? (being dramatic here since this is Nat. Comm. and not PRL).*

Reply: We thank the reviewer for pointing us to the need for a discussion of the broader implications of our work, which we are happy to provide. Besides the striking general relevance of the model organism in global ecology or the development of our atmosphere and climate, there is a number of implications of our observations regarding evolutionary, ecological, and engineering aspects. Nonetheless we would like to point out that it is not only the ecological or biotechnological relevance of our model organism that makes the work interesting for a general audience. Rather, the interdisciplinarity arises from the generality of the phenomenon: Alignment relative to external cues emerges for any elongated organism with responsive, bi-directional motility, which encircles a huge portion of the microbial domain.

Filamentous cyanobacteria exist with benthic and planktonic lifestyles. In both cases, filaments may disperse or aggregate. It is commonly assumed that aggregation offers an evolutionary advantage, e.g. for regulating exposure to environmental factors like illuminance [17]. Aggregation at light boundaries enhances the effect because the dark is easily accessible e.g. when excessive radiation levels induce phototoxic reactions.

In addition, aggregation requires individuals to encounter each other or existing aggregates. For simple organisms with bi-directional motility without active navigation, encounters are random and depend on density. Accumulation on a boundary significantly reduces the dimension of the search space and thus facilitates aggregation. Typically, freshwater filamentous cyanobacteria like *O. lutea* dwell on submersed soil, which naturally provides heterogeneous light conditions, so our observations align with the lifestyle of *O. lutea*. We are not aware of systematic studies of such structures in natural habitats, but this might be an interesting future research direction, as frequently pointed out in ecology literature [17].

We have rewritten the introduction and the discussion to include the broader relevance of the boundary accumulation phenomenon. This applies to the same text passages as the next comment, please find the modifications below.

Reviewer 2: *3. The same goes for the introduction and motivation for the work – it is weakly written. In the Intro, this sentence tries to discuss the gap in the literature, “While the statistical physics principles behind these emergent behaviors are well understood for simple, compact agents [31, 32], little is known about how they are affected by the strong shape anisotropy and flexibility of filamentous agents.” For example these ideas could be important for soft flexible robotics or roots etc – and a broader vision could really help put their work in a larger context.*

Reply: We thank the referee also for this comment on the context of our work. In the revision we have substantially revised introduction and discussion, to also include application aspects.

Engineered adaptive living materials are an emerging topic in bio-engineering research. Interestingly, the diameter of many filamentous cyanobacteria is comparable to that of carbon fibers, but they are motile and can be guided by light to self-assemble into yarns, braidings, or mats. Our work instructs how such yarns can be can be fabricated with various and, possibly, adaptive topologies, by simply projecting light patterns. We have rewritten the introduction and the discussion accordingly.

Introduction (page 1):

Emergent collective behaviors of motile biofilaments are ubiquitous across scales in nature, from cytoskeleton constituents [1–4] to colonies of elongated bacteria [5–10] and even millimeter-sized worms [11, 12], with promising applications in synthetic, bio-inspired materials [13–15]. In this context, filamentous cyanobacteria are currently attracting significant attention [9, 10, 16]. These microorganisms are indeed among the oldest, yet still most abundant

phototrophic prokaryotes on Earth, fixing vast amounts of atmospheric carbon by photosynthesis [17, 18]. They comprise a wide variety of phenotypes, habitats, and lifestyles, covering benthic and planctonic forms in fresh and salt water, as well as adaptations to extreme conditions like hypersaline lakes or hot springs [17]. Driven by climate change and eutrophication, harmful cyanobacterial blooms are intensifying in recent years [19, 20]. Such blooms typically develop when submersed populations grow to a critical density, upon which individual filaments aggregate and eventually rise to the surface of the water body [10, 17, 21]. The role of tychoplantc species, i.e. organisms that usually dwell on submersed surfaces but eventually aggregate, detach, and rise to the water surface, remains to be elucidated [17]. Beyond their major ecological impact [17, 22], cyanobacteria are gaining relevance as renewable energy source from photobioreactors or for biochemicals production [18].

Filamentous cyanobacteria are long and flexible [23, 24], containing up to several hundred, linearly stacked cells. Many species glide actively along solid surfaces or each other [10], exhibiting photo- and scotophobic responses [25] i.e., reversals of their direction of motion when they experience an increase or decrease in illuminance, respectively. These phenomena are currently not understood in detail [24, 26, 27], but appear to be crucial for aggregation, which, besides its role in blooming, is believed to provide light exposure regulation [28] and protection from other environmental factors [17]. Yet, the genesis of aggregation remains largely unknown [10], especially for benthic and tychoplantc species that are usually subject to spatially inhomogeneous illumination [9, 29]. In contrast to compact photoresponsive agents like *E. coli* [30, 31] or self-thermophoretic colloids [32], for which light-controlled aggregation is well studied [33, 34], filamentous agents generally entail additional emergent phenomena that are currently not well understood [10, 35].

Discussion (page 6):

... Since boundary accumulation leads to higher local densities, it enables colonies to form structures more efficiently with limited amounts of filaments. Such aggregates are believed to shield colonies from external influences while enabling individuals to modulate their exposure by moving through the aggregate [17]. Thus our observation aligns with the properties of the freshwater benthic habitat of *O. lutea* where strong light gradients e.g. from gravel or vegetation, are omnipresent.

Finally, we emphasize that, as the model does not include any specific detail about the biology of *O. lutea*, the collective effect we report here should be

observable in a wide variety of systems, enabling filamentous organisms with purely one-dimensional motility of re-orientation, alignment, and aggregation according to sensory cues from their environment. From an engineering perspective, defining nematic textures by simple light stimuli ultimately provides a tool to create biomimetic mesh materials with programmable topological structure.

Reviewer 2: *Minor Comments:*

1. *Is there any effect of light on the motion of cyanobacteria, it's not considered in the modeling.*

Reply:

This is indeed an interesting question because photokinesis in cyanobacteria has been reported in classical work [55], so filaments could move at different speeds in bright and dark regions. However, this is a weak effect, not present for all species. We could not detect significant photokinesis in our cultures of *O. lutea*. Filaments also continued to glide in complete darkness for ~ 48 h, much longer than the typical duration of the experiment. Therefore, we decided to not include photokinesis in the model.

To clarify this aspect, we added to the description of the accumulation experiments (page 2):

The response of the filamentous cyanobacterium *Oscillatoria lutea* to light gradients was observed with the experimental setup sketched in Fig. 1(a). *O. lutea* is a tychoplantonic freshwater species that forms sessile and floating colonies [36]. Individual filaments are negatively buoyant, but aggregates may develop gas bubbles, enclosed in a mesh of filaments and secreted slime, that cause flotation. The mechanical properties and collective behavior of *O. lutea* have recently been investigated [9, 23, 24, 37]. Filaments are about $5\ \mu\text{m}$ wide, flexible, and exhibit bi-directional gliding motility that is sustained in complete darkness. Filaments were contained in a Petri dish, ...

Reviewer 2: 2. *How will the shapes with inhomogeneous boundary curvature in Fig. 4 look for different sizes?*

Reply: We thank the reviewer for raising this question, which led us to perform an additional set of experiments and new simulations. As shown in the new appendix figure S7, boundary accumulation is lost for small patterns regardless of their shape, consistent with

the results shown in the main text for disks. This feature comes from the fact that filaments sitting along one edge of a polygon typically exhibit back and forth motion between two vertices. This type of motion naturally requires the typical filament length to be shorter than the edge, so that for sufficiently small polygons boundary accumulation is impossible.

Fig. S7: Density (first and third rows) and nematic order (second and fourth rows) profiles obtained from numerical simulations of the active polymer model for square and triangular illumination patterns. As the size of the pattern decreases, boundary accumulation is progressively lost. The scale bar is 20σ .

We performed two additional experiments with a smaller square and a smaller triangle, finding the same effect as in the simulations. These results are added to the manuscript as Fig. S8.

We also added a comment to the paper, referring to the new appendix figures (page 6):

Fig. S8: Snapshots from experiments with small square (top) and triangle (bottom) illuminations spots, with sizes comparable to the typical filaments length (scale bar: 1 mm). The last two snapshots (middle and right) are obtained shortly before and after the mask is removed after ~ 20 hours.

... This effect only breaks down when the edge length becomes comparable to or smaller than the typical filament length, similar to the case of disk-shaped regions. *Appendix* Figures **S7** and **S8** show densities and snapshots from simulations and experiments with smaller squares and triangles, respectively.

Our results demonstrate that filamentous cyanobacteria, which reverse their gliding direction in response to light gradients, exhibit emergent alignment ...

Reviewer 2: 3. Please give a step-by-step demonstration of nematic order estimation from images.

Reply: We thank the reviewer for requesting a detailed demonstration of the nematic order estimation, which gives us the opportunity to extend and clarify the respective paragraph in the *Methods* section, including the new figure **5** in the methods section of the main manuscript:

Nematic order parameter. An overview for the process of the nematic order estimation is given in Fig. **5**. To obtain a local orientation field, each frame is normalized by the maximum projection (see Fig. **5(b)**) and the negative logarithm is taken. Then, a morphologically eroded (square kernel of size 9×9 ,

Fig. 5: Nematic order estimation for a dense ring aggregate. (a) Raw micrograph of *O. lutea*, illuminated with a circular path of radius $R = 1.94$ mm after 10 h. (b) Raw micrograph divided by the maximum projection to compensate for varying intensities I across an image. (c) Local density variations obtained by subtracting a morphologically eroded (square kernel of size 9×9) version of a logarithmic image. (d) Local orientations in Gaussian-weighted neighborhoods. Depending on the intensity I , different filters (green color) are introduced for single filaments ($0.6 < I \leq 0.9$) and inside the dense ring ($0.1 < I \leq 0.6$). Combining both estimates yields the local orientation of the full micrograph, color indicates the orientation. (e) Norm of the nematic order tensor weighted by the sum of the two filters (single filaments + dense ring).

about twice the filament diameter) version of the logarithmic image is subtracted from the image itself, to obtain local density variations (see Fig. 5(c)). Convolutions with dedicated kernels then return the mean density $\langle w \rangle$, the local offset $\langle x \rangle$ and $\langle y \rangle$ of the center of density, and the expectations $\langle x^2 \rangle$, $\langle y^2 \rangle$, and $\langle xy \rangle$ of the local distribution of density N in Gaussian-weighted neighborhoods of each pixel. The local orientation is then calculated by

$$\Phi = \frac{1}{2} \arctan \left(\frac{-2 \left(\langle xy \rangle \langle w \rangle - \langle x \rangle \langle y \rangle \right)}{\langle x^2 \rangle \langle w \rangle - \langle x \rangle \langle x \rangle - \langle y^2 \rangle \langle w \rangle + \langle y \rangle \langle y \rangle} \right). \quad (1)$$

Here, the convolutions $\langle \cdot \rangle$ are taken with Gaussian-weighted, normalized kernels of 13×13 pixels (standard deviation = 6 pixels) where $0.6 < I \leq 0.9$ (single filaments) or 401×401 pixels (standard deviation = 200 pixels), where $0.1 < I \leq 0.6$ (dense ring). For regions with $I < 0.1$ or $I > 0.9$, the mean orientation is not taken into account. This is realized by introducing two filters for the

two different regions (single filaments and dense ring, see Fig. 5(d), left), where the green color indicates the regions where the condition for I is fulfilled. The convolutions are always taken for the full image shown in Fig. 5(c) and the filters are used to select the respective pixels. Both estimates are combined to yield the local orientation of the micrograph (Fig. 5(d), right). The components of the nematic order tensor are obtained by

$$Q_{xx} = \langle \cos^2(\Phi) - 1/2 \rangle, \quad Q_{xy} = \langle \cos(\Phi) \sin(\Phi) \rangle \quad (2)$$

and the norm of the nematic tensor is

$$Q = 2 \sqrt{Q_{xx}^2 + Q_{xy}^2}. \quad (3)$$

Here, the local average $\langle \cdot \rangle$ is taken with a normalized square kernel over 201×201 pixels to ensure ensemble averages that contain several filaments. The components Q_{xx} and Q_{xy} are multiplied inside the average by the sum of the two filters to take only values into account that fulfill $0.1 < I \leq 0.9$. For the plots in Fig. 2, the nematic order is averaged over the azimuthal angle to obtain the mean nematic order vs. r .

Reviewer 2: 4. *What is the finite probability of cyanobacteria moving into the dark region from the illuminated region used in the simulations? And how it is related to the experiment? Do they have similar probabilistic behavior in experiments?*

Reply: We thank the referee for this question which we clarified in the revised manuscript. In the experiments, isolated bacteria typically have a finite (typically small) probability to not reverse and escape the illuminated region. This probability is hard to quantify as it also depends on the history of exposure and the general condition of the filaments. For our experiments, we estimate the reversal probability to be at least 95%. As stated in methods, implementing this in simulations has no qualitative effect on the observed boundary accumulation (we tested cases where the reversal probability of filaments entering the dark region were 0.98 and 1). Hence, for all results presented in the main text filaments reverse their direction of motion with a probability 1 as they enter the dark region.

Reviewer 2: 5. *It would be interesting to look at how the cyanobacteria will aggregate when there are multiple illuminated regions in the dish.*

Reply: The referee certainly raises here a very interesting question, which opens the way

Fig. R3: Time-averaged filament density obtained from simulations with a pair of illuminated disks with radii 30σ placed at a relative center-to-center distance of $\Delta x = 60\sigma$. Both illuminated disk show stable and pronounced rings.

to the exploration of a whole set of more complex and disconnected geometries. Whereas the present work focuses on highlighting the emergent phenomenon of ring formation, the extension to multiple illuminated regions in future works is indeed natural. To satiate the reviewer’s curiosity, we present below preliminary numerical results for a configuration with two adjacent circular patterns. As shown in Fig. R3, two such patterns sufficiently far from each other both present rings and do not seem to strongly interact. The same can be observed on the experimental side in Fig. R4. Studying how the patterns start to affect each other as the patterns become closer and exchange filaments is certainly an interesting avenue which we intend to explore in the future.

Fig. R4: Two circular light patches ($R = 1.45$ mm) illuminated at the same time. Snapshots of the experiment after 0 h, 10 h, and 20 h are shown, scale bar is 1 mm. The field of view is limited by the lower end of the magnification range of our microscope setup.

Reviewer 2: 6. *The authors have shown the dissolution of cyanobacteria when the mask is removed. How does the cyanobacteria assembly evolve when the mask is shifted to create the illuminated region at a new position?*

Reply: We thank the reviewer for suggesting another interesting extension of our work. Since the current manuscript is focused on a new, emergent effect, we would like to exclude temporal variations from the present manuscript. Nonetheless, we think that this is a very interesting avenue for future work, which we have started to explore, so we provide here some first results.

Fig. R5 was obtained from an experiment where the center of a circular mask was shifted by one radius about a day after the beginning of the experiment. The existing structure dissolves and a new ring forms. However, a residue of the previous structure remains visible even after ~ 20 h, possibly related to the secretion of polysaccharides. As shown in Fig. R6, a qualitatively similar scenario is observed in our numerical simulations, however the previous structure is less persistent.

Fig. R5: Shifting the circular mask ($R = 1.94$ mm) after ~ 23 h by $1R$ to a new position. Time is given in upper right corner in the format hh:mm after start of the experiment.

Fig. R6: Simulation: Shifting the center of circular mask ($R = 40.$) after formation of ring by $R/2.$ to a new position.

Reviewer 2: 7. *What happens when the shapes are non-convex?*

Reply: We thank the reviewer for this question, which allows us to extend our findings to more general geometries. Fig. 4 of the main manuscript now includes an additional shape corresponding to a round corner concave square (panel (h)):

Fig. 4: Accumulation patterns for non-circular illumination. (a) Schematic of filaments that bend or splice into corners with obtuse or acute angles, respectively. The red arrows point towards the experimental observations of the trapezoid in (e). (b)–(h) Results from experiments and simulations. The left panel shows stationary configurations experimentally obtained after ~ 17 to 21 hours of illumination (scale bar is 1 mm). The right panel corresponds to the orientation of the time-averaged nematic order in numerical simulations of the active filament model (scale bar is 20σ). The saturation has been adapted to reflect the time averaged density, so that dilute regions appear darker. The color code for the orientation is given in panel (b), while the winding number corresponding to each shape (see main text) appears at the top left corner.

Both simulations and experiments reveal that boundary accumulation remains robust to mild locally negative interface curvatures, which further highlights the robustness of this phenomenon. We rewrote the corresponding paragraph in the manuscript (page 5):

...near the strongly curved apices of the ellipse. These characteristics also remain robust for non-convex shapes so long as they present sufficiently low curvature, as illustrated in Fig. 4(h) for a round corner concave square. In contrast, the presence of regions with curvature exceeding the threshold for ring formation (Fig. 2(d)) results in a qualitative change of the filament organization. For example, for an ellipse with strong eccentricity such as the one shown in Fig. 4(c), the filaments no longer bend to follow the interface, but rather splice

into it, thus locally forming a splay deformation pattern (see Fig. 4(a) for a sketch of the deformation patterns and *Appendix Fig. S6* for local orientation of the experimental snapshots). Remarkably, this transition

Additionally, we updated Fig. **S6** in the *Appendix* accordingly.

Fig. S6: Local orientation for non-circular illumination. (a)–(g) show the local orientation of stationary configurations experimentally obtained after ~ 17 to 21 hours of illumination, calculated with Eq. (4). The color code for the orientation is given in panel (a), scale bar is 1 mm.

Reviewer 2: 8. *Can there be some application of these findings? Or can it be connected to some assemblies already seen in nature?*

Reply: Applications and natural occurrence of the phenomenon is indeed an interesting point. This request has been addressed in the replies to requests 2 and 3, asking for the broader context of the phenomenon. Applications and natural assemblies are now addressed in the re-written introduction and discussion. Please see above for the detailed changes.

Reviewer 2: 9. *Will the overlapping of the cyanobacteria change when we have more of them in the petri dish? How will their aggregation change?*

Reply: Indeed, this is an interesting point. Aggregation requires a critical filament density, which is a general phenomenon for active filaments and has recently also been

explored for filamentous cyanobacteria [9]. Due to scotophobic reversals, the density in the light spot gradually increases over time as dispersed filaments randomly encounter the light spot. Therefore, the primary effect of the seeding density is to set the time scale after which boundary aggregation starts. Seeding density in the experiments naturally varied, so the onset ring formation also varied. This can be seen in supplementary figure. **S2**, where the density in the periphery of the light spot (i.e., the region where the ring forms) is plotted against the density in the central region, showing time as color code. Of course, a critical minimum seeding density is required to observe the onset of aggregation on a reasonable time scale. Likewise, the seeding density should be low enough to exclude spontaneous aggregation since we would like to isolate the effect of the light boundary. In between, there is a wide range of seeding densities at which boundary accumulation can robustly be observed.

Reviewer 2: *10. How is the self-propulsion force chosen in simulations? Does it change with time? How does it correlate with the experiments?*

Reply: As the cyanobacteria remain motile throughout the experiment (cf. Supplemental Movie 2 when the mask is removed), the self-propulsion force is independent of time in the simulations. As we also argued in our response to Reviewer 3, the goal of our modeling is not to offer a quantitative match with the experiment, but to highlight the generality of the collective behavior we observe and identify the minimal set of physical ingredients needed for its emergence. In fact, our modeling demonstrates the main requirement is that the self-propulsion force is sufficiently large to allow the filaments to cross each other. If this condition is not satisfied, filaments spontaneously form polar clusters that destabilize the order and prevent any steady structure formation, as discussed in the section “Simulations and emergent nature of the alignment” of the main text, and shown in Supplemental Movie 4.

Reviewer 2: *11. The word “scotophobic” is defined on the last page – this should be up front and center – this is not a common term and is central for the reader to appreciate these findings – burying it in the conclusion does not help the author’s case.*

Reply: Thank you very much for pointing this out —we have shifted the definition to the introduction (page 1):

...Many species glide actively along solid surfaces or each other [10], exhibiting photo- and scotophobic responses [25] i.e., reversals of their direction

of motion when they experience an increase or decrease in illuminance, respectively. These phenomena are currently not understood . . .

Reviewer 2: 12. *Why the winding number (Eq. 2) is discussed in detail is not very clear – is it really important for the paper? What is the point of it?*

Reply: We thank the reviewer for giving us the opportunity to make this point of the manuscript clearer. Particularly, we show how boundary accumulation can be used to sculpt the morphology of scotophobic filament ensembles in a systematic and reproducible way. Topology plays an important role in the mechanical properties of yarns and (semi-random) meshes. The winding number then provides a topological measure to characterize the properties of the pattern (i.e. the relative amounts of bend and splay deformation). This appears particularly clearly in the panels (b) and (c) of Fig. 4, which show that the transition from a bend to a splay deformation when the local curvature at the apices of the ellipse exceeds a given threshold is associated with a change in the winding number of the overall pattern. In the case of convex polygons, Eq. (2) then directly relates this observable to the geometry of the illumination pattern.

As mentioned above regarding applications, self-assembling filaments into meshes, yarns or braidings offers intriguing approaches for designing living fiber-based materials. Controlling the topology of the assembly is important because different patterns exhibit different properties. For instance, aggregates cannot be bent below a certain, critical radius, switching to a splay pattern instead where no fibers are aligned in the direction of hoop stresses. Therefore, the induction of topological charge in active fibrous materials is intensely discussed in fundamental science literature.

Reply to Reviewer #3

Reviewer 3: *In this article by Kurjahn et al, the authors pair experiments and simulations to look at ‘scotophobic’ i. e. darkness-avoiding filamentous cyanobacteria. These long, thin filaments reverse gliding direction when coming into contact with darkness. The authors take advantage of this by constructing round and polygonally shaped patterned regions of illumination. The filaments then accumulate tangentially to the boundaries, forming a ring for circular regions and patterns that include discontinuities for polygonal shapes.*

The authors provide a convincing semi-quantitative explanation of the filament orientation and density patterns by pointing out that filament motion patterns will cause them to accumulate along secants to the circle. They then pair experiment with a qualitative active polymer simulation to compute nematic orientation, density profiles and the patterns for systems with sharp corners. For the latter, they propose a counting rule for the winding number, $w = 1 - n_a/2$ that counts the number of acute angles n_a where the filaments have what looks like a splay defect to me.

Reply: We thank the referee for this precise and positive summary of our work. Please find below our detailed response to each comment.

Reviewer 3: *The accompanying simulation is qualitative, with no real attempt at matching the experiment quantitatively:*

We would like to emphasize here that the goal of our modeling is indeed not to produce a numerical twin of the experiment—which may be very useful in some cases—but rather to highlight the minimal physical ingredients needed to reproduce a nontrivial emergent behavior. It is then remarkable that we can qualitatively (and even semi-quantitatively, see e.g. Fig. 2(d) of the main text) reproduce the experimental observations without needing to fine tune or fit the model parameters. This, we believe, greatly enhances the scope of our findings as we expect them to be very generic and robust, and therefore to apply to a wide variety of biological and synthetic systems and not only cyanobacteria. This point is now further emphasized in the discussion.

– *Why is this not in fully reduced units? The choice of units is related to a thermal monomer diffusion time which looks irrelevant here. The natural choice would be using a unit friction coefficient, particle size and then one more number to set a force scale, e. g. the active force magnitude. Instead, there are what seems to be a bunch of arbitrary*

numbers with no attempt at understanding scales.

Reply: We thank the reviewer for pointing out this shortcoming in the presentation of the numerical model. Indeed, we agree that using σ and the typical self-propulsion velocity to fix the units of length and time in the model is more natural. Therefore, we have modified the presentation of the model in Methods in order to account for this new choice, as can be seen in the text below.

All simulations were performed in a two-dimensional periodic box of size 188σ . To account for the polydispersity of the filament lengths observed in the experiments (Fig. S1), in our simulations the filament lengths were drawn from a Poisson distribution with a fixed mean 20σ . As the number of filaments was fixed to $N_f = 1020$, the simulated systems comprise on average $\approx 20\,000$ monomers. Choosing σ and $\sigma\zeta/f_{\text{act}}$ as units of length and time, respectively, the dimensionless Eq. (7) depends on five non-dimensional parameters, namely

$$\tilde{\kappa}_s = \frac{\kappa_s\sigma}{f_{\text{act}}}, \quad \tilde{\kappa}_b = \frac{\kappa_b}{\sigma f_{\text{act}}}, \quad \tilde{U}_0 = \frac{U_0}{\sigma f_{\text{act}}}, \quad \tilde{D} = \frac{k_B T}{\sigma f_{\text{act}}}, \quad \tilde{d} = \frac{d}{\sigma}.$$

The parameter values chosen in simulations are reported in Table 1. The equations of motion (7) were integrated with a time step $dt = 10^{-3}$ for typically 10^7 simulation steps, which we checked was much larger than needed for the dynamics to reach a steady state. The large value of $\tilde{\kappa}_s = 500$ reported in Table 1 ensures marginal stretching of the filaments. We moreover checked that the results reported in the main text do not depend on any fine tuning of the model parameters, as they remain qualitatively unchanged when, e.g., the filament stiffness κ_b or the self-propulsion force f_{act} are changed.

Table 1: Simulation parameters corresponding to the data shown in the main text (unless specified otherwise).

Parameter	$\tilde{\kappa}_s$	$\tilde{\kappa}_b$	\tilde{U}_0	\tilde{d}	R/σ	\tilde{D}
Value	500	5	2.4	0.2	60	$5 \cdot 10^{-6}$

Reviewer 3: – *I would have expected the relevant scaling for the secant density profiles to be the persistence length ξ of the filament scaled by the radius R of the circle, or if very stiff, its length (from the movie, I can just about see that the filament is nearly but not totally stiff). There is no discussion about any of this.*

Reply: The impact of the persistence length is a relevant point. Indeed we found

that the typical secant length is comparable to the filament length (p.4 after eq. 1), corresponding to “very stiff” filaments, as pointed out also by the referee. How is that possible, even though filaments are flexible enough such they visibly bend in the movies?

First of all, the thermodynamic persistence length is $\xi = B/(kT)$, where B is the bending modulus of the filament. Recent measurements [23, 24] indicate that B is on the order of 10^{-17} to 10^{-16} Jm, so $\xi \sim 10^4$ m, eight orders of magnitude larger than the typical filament length [23]. Obviously, bending must be either spontaneous or related to the self-propulsion forces and the mechanical interaction of the filaments. Filaments sometimes exhibit spontaneous curvature on the order of several millimeters [9], depending on culture conditions, but these are larger than our light patterns.

There is no simple estimate for the order of magnitude of random mechanical forces in ensembles of active filaments because these depend on the emergent structure in the ensemble. One estimate could be the critical self-buckling length [24], which is on the order of a few hundred microns, indeed relevant for the secant estimation. However, this corresponds to a complete stall of the motion by hitting a mechanical obstacle. In our experiments, there is not physical wall which could induce bending: the confinement is purely virtual. Thus, filaments could only bend by exerting forces onto each other. The length scale of such spontaneously emerging structures is typically larger than our light patterns [9].

Thus we conclude that bending remains negligible, mainly because the confinement is virtual: entering the dark does not exert any force on the filament. We included this discussion on the absence of significant bending in the discussion of Eq. (1) on page 4 of the manuscript:

... These findings suggest that bending into curved boundaries remains limited, leading to the vanishing of boundary accumulation at a finite domain radius. **The absence of bending is a consequence of the virtual confinement: Although self-propulsion forces are sufficient to induce significant bending [24], there is no physical wall to counter the thrust. Spontaneous curvature or bending due to thermal or active fluctuations [9, 23] are smaller than the structures we investigate here.**

Reviewer 3: – *The relatively weak interactions between filaments are modelled here by a soft harmonic potential with range d and strength U_0 . This accounts for filaments that can overlap. There is a transition towards filament alignment with d here, but U_0 is*

a fixed number. I would have kept d fixed and then expected that $U_0 d/f_{act}$ would act as a scale of the alignment strength.

Reply: The reviewer is correct as both U_0 and d can in principle be varied to tune the strength of the interaction. This point is discussed in more detail in the Methods section, which we have revised to highlight further the reasons which led to the choice of varying d and not U_0 :

For sufficiently large U_0 and d , filaments experience strong repulsive forces that prevent overlaps and thus reproduce a strictly 2D confinement. Decreasing either of these two parameters, we observe crossing between filaments similar to those reported in the experiments (see Fig. 1(d)). Lowering U_0 at fixed $d = \sigma/2$ for the filament density used in our simulations, this transition quickly leads to a vanishing of the interactions between filaments, leading to structureless aggregates in the illuminated region. On the other hand, decreasing d at fixed U_0 allows for a larger intermediate regime where filaments can cross each other while still effectively aligning nematically when their density reaches large enough values. As discussed in the main text and shown in Fig. 3, this regime where $0.15 \lesssim d/\sigma \lesssim 0.35$ precisely corresponds to the emergence of nontrivial patterns at the illumination boundary.

Namely, contrary to previous models of active filaments (see e.g. Refs [2, 9, 48]), we do not implement explicit aligning interactions but only pure repulsion. We must therefore work in an intermediate regime where interactions are sufficiently strong to ensure alignment, but weak enough so that filaments can cross. As stated above, we have tried to vary both U_0 and d in order to mimic this effect. For the parameters chosen, we have found that fixing $d = \sigma/2$ and decreasing U_0 typically led interactions too weak to promote alignment, which is why we chose instead to fix U_0 and vary d . Note that this effect is likely to depend on the global filament density. Considering higher densities would probably allow to recover a higher degree of alignment, even for weak U_0 , but at the cost of numerical efficiency.

Reviewer 3: *For the non-circular confined systems: How can the sharp corner and winding number calculation be linked to the usual language of topological defects? All of the systems shown here are isomorphic to a circle with Euler number 1. Then with a tangent boundary condition, I would expect the sum of the charges of the topological defects in the system to always be 1 as well. Of course, the corners may introduce issues here, and giving that filaments can cross local nematic order may be hard to define everywhere. I tried to construct the following defect patterns, but it is not quite satisfactory:*

Reply: We highly appreciate this in-depth comment, which highlights several important features of the emergent alignment mechanism. Indeed, the outline of all these shapes correspond to a topological charge of $+1$ which is why the counting rule contains an offset of 1, as pointed out by the referee. The subtlety arises in the alignment of the filaments: there is no strict boundary condition regarding the orientation of the filaments, since there is no physical confinement and the alignment is an emergent effect. In other words, there is no anchoring condition. Therefore, the winding number of the boundary accumulation structure is not identical to the winding number of the shape, caused by the switch from bend to splay as angles become acute or the local curvature of the interface is too high.

A similar phenomenon has been reported for dense nematics in several works. In particular, as shown in Ref. [50] (see Figure 3 in that reference) the defect pattern induced by confinement in nematic liquid crystals depends on the type of deformation induced by the vertices. Geometrically, continuously mapping the splay deformation at a corner to a circle retains a singular point with vertical alignment (see sketch Fig. R7 below), so that the total charge of the nematic texture is not constrained by the Euler characteristic of the confinement pattern.

Fig. R7: Top: a confined nematic pattern in a polygon with bend deformation at the vertices continuously maps to a disk pattern with tangential alignment. Bottom: when the polygonal pattern exhibits splay deformation at the vertices, it can only be continuously deformed into a disk pattern with nonuniform boundary condition, resulting in a different topological charge. Red and blue triangles indicate negatively and positively-charged topological defects, respectively.

In order to make these considerations more clear, Fig. 4 of the main text in this document) now includes in panel (a) a sketch of the bend and splay deformation patterns found respectively at obtuse and acute angles:

Fig. 4: Accumulation patterns for non-circular illumination. (a) Schematic of filaments that bend or splice into corners with obtuse or acute angles, respectively. The red arrows point towards the experimental observations of the trapezoid in (e). (b)–(h) Results from experiments and simulations. The left panel shows stationary configurations experimentally obtained after ~ 17 to 21 hours of illumination (scale bar is 1 mm). The right panel corresponds to the orientation of the time-averaged nematic order in numerical simulations of the active filament model (scale bar is 20σ). The saturation has been adapted to reflect the time averaged density, so that dilute regions appear darker. The color code for the orientation is given in panel (b), while the winding number corresponding to each shape (see main text) appears at the top left corner.

The corresponding discussion in the main text was modified to include additional explanations (page 6):

... filaments splice into the corner. The resulting splay deformation pattern then leads to a sudden reversal of the direction of rotation of the nematic order along the mask edge (Fig. 4(a)). In both simulations and experiments, the corner angle at which the filament deformation transitions from bend to splay is found to be about 90° .

To quantitatively relate the geometry of the virtual confinement to the morphology of the bacterial pattern, we consider the winding number associated with the circulation of the filament orientation along the boundary of the illuminated region. Since a bend (splay) deformation corresponds to a rotation of the filament pattern by an angle $\pi - \alpha$ ($-\alpha$) at a vertex with opening angle α (Fig 4(a)), simple geometric arguments predict that the total rotation along a closed curve defining the boundary is given by $2\pi w$, where the winding number is

$$w = 1 - \frac{n_a}{2}. \quad (2)$$

Here, n_a stands for the number of acute angles in the polygon (details in *Methods*). As shown in Fig. 4, Equation (2) is verified for all shapes previously described, so that the structure of the colony is fully determined by the geometry of the illumination pattern. Namely, in the absence of acute angles $w = 1$ (Figs. 1(c,d) and 4(b,d)), while for the flattened ellipse of Fig. 4(c) the two apices with high local curvature behave similarly to acute angles, leading to $w = 0$. A zero winding number is also obtained for trapezoids (Fig. 4(e)), while equilateral triangles and teardrop-like shapes give rise to $w = -\frac{1}{2}$ and $w = +\frac{1}{2}$, respectively (Figs. 4(f,g)). Note that, as convex polygons satisfy $n_a \leq 3$, the shapes displayed in Fig. 4 cover all possible values of w .

The above features are reminiscent of two-dimensional nematic liquid crystals, for which similar topological transitions between different frustrated defect patterns are induced by geometrical confinement [50]. Here, however, the bacteria are not subject to any physical boundary, but the mechanical confinement is self-generated by their emergent accumulation at the edges of the pattern. This effect only breaks down when the edge length becomes comparable to or smaller than the typical filament length, similar to the case of disk-shaped regions. *Appendix* Figures S7 and S8 show densities and snapshots from simulations and experiments with smaller squares and triangles, respectively.

Details on the derivation of Eq. (2) were added to Methods (page 9):

Nematic patterns in convex polygons. As described in the main text, boundary accumulation induced by convex light patterns is characterized by a bending (spicing) of the filaments at vertices with obtuse (acute) angles. To evaluate the total circulation of the filament orientation along the edge of a polygon, we note that inside a corner with opening angle α , the filament orientation in the boundary structure may change either by $\pi - \alpha$ or by $-\alpha$, corresponding to bend and splay, respectively (see Fig. 4(a)). Considering a convex polygon with n_a acute and n_o obtuse angles, the winding number is thus given by

$$w = \frac{1}{2\pi} \left[\sum_{i=1}^{n_a} (-\alpha_i) + \sum_{j=1}^{n_o} (\pi - \alpha_j) \right].$$

Noting that the angles of the polygon satisfy $\sum_{i=1}^{n_a} \alpha_i + \sum_{j=1}^{n_o} \alpha_j = (n_a + n_o - 2)\pi$, it is then straightforward to recover Eq. (2) of the main text.

Reviewer 3: *Finally, the structure of the article is a bit strange. There is a long paragraph about the simulations in the initial ‘Patterned illumination’ section, starting with ‘To confirm that the formation of the ring’. This feels out of place.*

Reply: We thank the referee for pointing out this issue with the article structure. Since we compare experimental and numerical results side-by-side, we would like to introduce the basic ingredients of the simulations early on. Apart from some minor rewordings, we have shifted the model part by one paragraph and modified the section headings to reflect the content better (“Accumulation at light boundaries”, “Minimal model & mechanism”, and “Filament crossings & three-dimensional structure”).

Reviewer 3: *One more minor point: Please define ‘scotophobic’ the first time you use it.*

Reply: Thank you very much for pointing this out – we have shifted the definition to the introduction (page 1):

... Many species glide actively along solid surfaces or each other [10], exhibiting photo- and scotophobic responses [25] i.e., reversals of their direction of motion when they experience an increase or decrease in illuminance, respectively. These phenomena are currently not understood ... These phenomena are currently not understood ...

Reviewer 3: *All in all this is a nice experiment, but I am not sure if in its current state the level of model and explanation is quite sufficient for a high profile publication.*

Reply: We thank the referee again for the positive evaluation of our work. We have substantially amended the simulations part. Still, the simulations are intended to determine a minimal set of ingredients from which the underlying physics principles of the phenomenon can be inferred. A full-detail quantitative model would certainly be interesting for various other objectives, but rather complicate that reductionist approach which we consider important for a first understanding of a new phenomenon.

REVIEWERS' COMMENTS

Reviewer #1 (Remarks to the Author):

The authors responded well to my comments and adjusted the presentation to include a more ecology-oriented audience.

A final minor (and optional) suggestion is to include Fig. R2, which I found very helpful, as one of the supplementary figures.

I recommend publication in Nature Communications.

Reviewer #2 (Remarks to the Author):

The authors have sufficiently addressed all my comments

I approve publication.

Congrats

Reviewer #2 (Remarks on code availability):

Ideally code should be shared via a permanent doi - GitHub?

Reviewer #3 (Remarks to the Author):

I would like to thank the authors for their in-dept response to all referee comments, and I wanted to apologise for the overly critical tone of my first report. The revisions strengthen the manuscript, highlighting both the interdisciplinary relevance of the research, and making the "PRL like" physics part more precise.

In particular, using rescaled units for the simulations makes it much clearer what is happening. I think the authors misunderstood my 'bending' comment slightly - it seems clear that the experimental filaments bend little or not at all. For the simulated ones, with a scaled bending stiffness of 5, some bending should be expected and is indeed visible in the simulations. Nonetheless, as the results appear robust to varying these parameters and reproduce the experiment, this appears fine.

I also appreciate the motivation for of varying d instead of U_0 to obtain a broader range of overlapping simulations.

I appreciate the in-depth discussion of the link to nematics in confinement (or the lack thereof). The point about virtual confinement not being boundaries is well made. The additional Figure 5a and especially the more in-depth derivation of equation 2 in the methods are appreciated.

Finally, making the data and also the simulation code publicly available is great and should become standard.

I recommend this article for publication in its current form.

Reviewer #3 (Remarks on code availability):

I downloaded the code and managed to compile it. However, I then received a segmentation fault when attempting to execute it.

Unfortunately, there is no readme file for installation and running the simulation, as I am certain the issue is along the lines of not reading the correct initial files.

Reviewer #4 (Remarks to the Author):
